# PASK links cellular energy metabolism with a mitotic self-renewal network to establish differentiation competence

**Michael Xiao[1,2†], Chia-Hua Wu[1†], Graham Meek[1†], Brian Kelly[1], Dara Buendia Castillo[1], Lyndsay EA Young[3], Sara Martire[4], Sajina Dhungel[1], Elizabeth McCauley[1], Purbita Saha[4], Altair L Dube[4], Matthew S Gentry[3], Laura A Banaszynski[4], Ramon C Sun[3,5], Chintan K Kikani[1]\***

[1]Department of Biology, College of Arts and Sciences, University of Kentucky, Thomas Hunt Morgan Building, Lexington, United States; [2]Weill Cornell/Rockefeller/ Sloan Kettering Tri-Institutional MD-PhD Program, New York, United States; [3]Molecular and Cellular Biochemistry, College of Medicine, University of Kentucky, Lexington, United States; [4]Cecil H. and Ida Green Center for Reproductive Biology Sciences, Children's Medical Center Research Institute, Department of Obstetrics & Gynecology, Hamon Center for Regenerative Science and Medicine at the University of Texas Southwestern Medical Center, Dallas, United States; [5]Department of Neuroscience, College of Medicine, University of Kentucky, Lexington, United States

**\*For correspondence:**
chintan.kikani@uky.edu

†These authors contributed equally to this work

**Competing interest:** The authors declare that no competing interests exist.

**Abstract** Quiescent stem cells are activated in response to a mechanical or chemical injury to their tissue niche. Activated cells rapidly generate a heterogeneous progenitor population that regenerates the damaged tissues. While the transcriptional cadence that generates heterogeneity is known, the metabolic pathways influencing the transcriptional machinery to establish a heterogeneous progenitor population remains unclear. Here, we describe a novel pathway downstream of mitochondrial glutamine metabolism that confers stem cell heterogeneity and establishes differentiation competence by countering post-mitotic self-renewal machinery. We discovered that mitochondrial glutamine metabolism induces CBP/EP300-dependent acetylation of stem cell-specific kinase, PAS domain-containing kinase (PASK), resulting in its release from cytoplasmic granules and subsequent nuclear migration. In the nucleus, PASK catalytically outcompetes mitotic WDR5-anaphase-promoting complex/cyclosome (APC/C) interaction resulting in the loss of post-mitotic Pax7 expression and exit from self-renewal. In concordance with these findings, genetic or pharmacological inhibition of PASK or glutamine metabolism upregulated Pax7 expression, reduced stem cell heterogeneity, and blocked myogenesis in vitro and muscle regeneration in mice. These results explain a mechanism whereby stem cells co-opt the proliferative functions of glutamine metabolism to generate transcriptional heterogeneity and establish differentiation competence by countering the mitotic self-renewal network via nuclear PASK.

## Editor's evaluation

The study by Xiao et al. presents an important finding in the area of metabolic regulation underpinning cell fate decisions in murine muscle stem cells. Combining multiple approaches, the study provides convincing evidence of glutamine-dependent control of the sub-cellular localisation of the kinase PASK and the consequent activation of myogenic programs. The study will be of interest to researchers in the areas of stem cells, regeneration, and metabolic signalling.

## Introduction

Cell identity is a dynamic feature that must be continually reestablished in proliferating stem cells. To preserve stem cell identity, self-renewing stem cells rapidly reactivate the expression of genes linked with cell identity following cell division. Several mechanisms have been proposed for the post-mitotic reactivation of lineage-defining genes, including the mitotic recruitment of the E3 ubiquitin ligase anaphase-promoting complex/cyclosome (APC/C) by WDR5 to transcriptional start sites of genes regulating pluripotency and H3K27ac mitotic bookmarking (*Pelham-Webb et al., 2021*; *Liu et al., 2017*; *Oh et al., 2020*). During mitosis, the WDR5-APC/C interaction is enhanced, and transient disruption of this interaction causes loss of pluripotency in embryonic stem cells (ESCs) (*Oh et al., 2020*). However, how differentiation signals counter mitotic self-renewal machinery remains poorly understood.

Expression of PAS domain-containing kinase (PASK), a kinase involved in cellular energy balance and metabolic control, positively correlates with the undifferentiated, proliferative state of ESCs and adult stem cells. Functionally, PASK is required for the onset of ESC and adult stem cell differentiation programs downstream of nutrient signaling (*Kikani et al., 2019*; *Kikani et al., 2016*). In adult muscles, PASK is required to induce an early regenerative myogenesis program (*Kikani et al., 2016*). During muscle regeneration, normally quiescent Pax7+ muscle stem cells (MuSCs) enter the cell cycle and undergo proliferative bursts, resulting in the generation of a heterogeneous, activated, self-renewing myoblast population (Pax7$^{+/-}$, MyoD$^{+/-}$, Myf5$^{+/-}$). Signaling cues stimulate the generation of a committed progenitor population (MyoD$^+$, MyoG$^+$), which orchestrates the myogenesis program. We and others have shown that PASK is required to generate the MyoD$^+$, MyoG$^+$ committed progenitor population during myogenesis. Mechanistically, PASK phosphorylates WDR5, a member of the histone H3 lysine 4 methyltransferase (H3K4me3) complexes, in response to differentiation signaling. This phosphorylation sets in motion chromatin remodeling at the *Myog* promoter and its transcriptional activation resulting in the onset of myogenesis (*Kikani et al., 2019*; *Kikani et al., 2016*; *Karakkat et al., 2019*). This entire pathway is downstream of nutrient-activated mTOR-dependent phosphorylation of PASK, which stimulates the PASK-WDR5 interaction (*Kikani et al., 2019*). Intriguingly, the PASK function is specifically required to establish the initial committed myoblast progenitor population but is dispensable to sustain myogenesis (*Kikani et al., 2019*). Since PASK functions at the critical decision point between self-renewal and differentiation, understanding signals that regulate PASK activity and subcellular distribution could provide mechanistic insight into how the exit from self-renewal is regulated in stem cells to generate the committed progenitor population.

Signaling cues from the early regenerating niche play critical regulatory roles in facilitating the transition from quiescent, non-proliferative MuSCs to activated, hyper-proliferative myoblasts during the early stages of tissue regeneration (*Ryall et al., 2015*). Mitochondrial uptake of glutamine is thought to play an essential role in sustaining stem cell proliferation by generating ATP and maintaining redox balance (*Yu et al., 2019*). During muscle regeneration, glutamine secreted by macrophages sustains myoblast proliferation and promotes differentiation (*Shang et al., 2020*); however, the precise mechanistic function of glutamine metabolism in driving myoblast differentiation is unclear. *Our* results discovered a novel mitochondrial-nuclear signaling axis that connects glutamine metabolism in the mitochondria with the mitotic self-renewal network in the nucleus. In this pathway, glutamine metabolic signaling is required to generate a differentiation-primed progenitor population by disrupting the cell cycle-linked WDR5-APC/C interaction via PASK acetylation and nuclear localization. This axis provides new insights into the regulation of stem cell self-renewal and differentiation and identifies key signaling pathways that could be targeted to enhance tissue regeneration.

## Results

### PASK inhibition preserves self-renewal and sustains the proliferation of adult stem cells and ESCs

PASK is highly expressed in proliferating pluripotent, embryonic, and adult stem cells from mice and humans (*Kikani et al., 2016*). It is rapidly downregulated following the onset of the differentiation program in all systems resulting in the near absence of PASK expression in most adult tissues under normal physiology (*Kikani et al., 2016*). Functionally, PASK is required for the onset of terminal differentiation program in embryonic and adult mouse stem cells; however, the role of PASK in self-renewal

and stemness properties of stem cells remains unclear. To answer these questions, we cultured mouse embryonic stem cells (mESCs) in the 2i+LIF (2i) condition designed to maintain pluripotency and subsequently replaced the 2i media with PASKi (PASK inhibitor, BioE-1197, *Kikani et al., 2016*; *Wu et al., 2014*)+LIF (PASKi) to assess if PASKi can sustain pluripotency after the withdrawal of 2i. Strikingly, replacing 2i media with PASKi in mESCs resulted in a further increase in the expression of genes associated with self-renewal and stemness (*Pou5f1, Sox2,* and *Prex1* mRNA) when compared with mESCs cultured in the 2i conditions (*Figure 1A*, *Figure 1—figure supplement 1A*). Additionally, using a Rex1-GFP reporter mESC line (*Wray et al., 2011*), we observed that cells cultured in PASKi maintained GFP reporter expression at levels comparable to those cultured in 2i (*Figure 1B*, *Figure 1—figure supplement 1B*). To compare the differentiation competence of 2i vs. PASKi cultured mESCs, we performed an embryoid body (EB) formation assay of cells grown in 2i or PASKi. Cells grown in the 2i culture condition differentiate well, as seen from the emergence of several fluid-filled cavitated structures after the withdrawal of 2i. Remarkably, PASKi cultured cells showed a substantial increase in the numbers and size of fluid-filled cavitated structures compared with 2i pretreated cells (~52% for PASKi versus ~12% for 2i-treated cells, *Figure 1—figure supplement 1C*) after PASKi withdrawal. Consistent with our previous study, the presence of PASKi during EB formation attenuated differentiation as assessed morphologically (*Figure 1—figure supplement 1C*; *Kikani et al., 2016*). Thus, PASK inhibition in ESCs preserved self-renewal and enhanced differentiation potential upon PASK inhibitor withdrawal.

PASK is highly expressed in adult proliferating stem cells (*Kikani et al., 2016*). Adult MuSCs undergo successive transitions through quiescence, activation, commitment, and differentiation during regeneration in vivo and in culture upon their isolation. While PASK is expressed at negligible levels in adult quiescent MuSCs, its expression increases rapidly as MuSCs are activated and begin to proliferate (*Figure 1—figure supplement 2A*; *Kikani et al., 2016*; *Liu et al., 2013*). To test if PASK inhibition during in vitro activation of MuSCs affects self-renewal, stemness, and differentiation dynamics of adult stem cells, we isolated primary myoblasts from hindlimbs of uninjured mice using flow cytometry-based sorting of Sca-1⁻, CD31⁻, CD45⁻, VCAM1+, α7-Integrin+ cells and cultured them in the presence or absence of PASKi. PASK inhibition was well tolerated by isolated primary myoblasts and caused no apparent proliferation defect. On the other hand, PASKi robustly blocked the precocious myogenesis observed in the control myoblasts for as long as 4 days post-isolation (*Figure 1C*, *Figure 1—figure supplement 2B–D*). Taken together, these results suggest that PASK inhibition preserves the self-renewal property of cultured ESCs and adult MuSCs and prevents their precocious differentiation.

To mechanistically understand how PASK functions during the onset of myoblast differentiation, we isolated primary myoblasts from *Pask^WT* and *Pask^KO* animals using fluorescence-activated cell sorting (FACS)-based purification (*Figure 1—figure supplement 3A*). MuSCs isolated from *Pask^WT* or *Pask^KO* were indistinguishable in size 12 hr post-isolation and did not show overt proliferation defects, which is consistent with our previous results and the notion that PASK is not expressed or required for the maintenance or release from a quiescent state. Finally, while *Pask^WT* cells began to form nascent myotubes by 48 hr post-isolation, *Pask^KO* cells continued to proliferate and showed little signs of myotube formation (*Figure 1—figure supplement 3B*). Thus, genetic or pharmacological loss of PASK results in continued self-renewal of adult myoblasts and impaired differentiation.

Quiescent MuSCs express Pax7 but lack MyoD or Myf5 protein expression (*Figure 1—figure supplement 2A*). Upon their isolation, Pax7+ MuSCs rapidly activate MyoD mRNA translation, and by 48 hr, Pax7+/MyoD+ SCs diverge into a heterogeneous population expressing a combination of Pax7, MyoD, and/or Myf5 (*Figure 1D–E*; *Rocheteau et al., 2012*). Interestingly, *Pask^KO* MuSCs showed an increased percentage of Pax7⁺ myoblasts numbers compared with control (~83% for *Pask^KO* vs. 30% for *Pask^WT*) during 48 hr of culture. Similarly, loss of PASK resulted in reduced levels of MyoD+ myoblasts (*Figure 1E–F*). Furthermore, the increased proportion of Pax7+ myoblasts observed in *Pask^KO* is independent of the method chosen for stem cell isolation (FACS, magnetic-activated cell sorting [MACS], and pronase-based method) (*Figure 1—figure supplement 3C–D*). Finally, *Pask^KO* myoblasts showed a marked reduction in committed MyoG-expressing cells and myogenesis, as measured by the fusion index (*Figure 1E–G*). Thus, genetic loss of PASK increases self-renewing Pax7+ myoblast numbers and decreases the generation of committed (MyoD+/MyoG+) myoblasts in vitro. During adult muscle regeneration, Pax7+ myoblasts begin to proliferate and generate MyoD+/

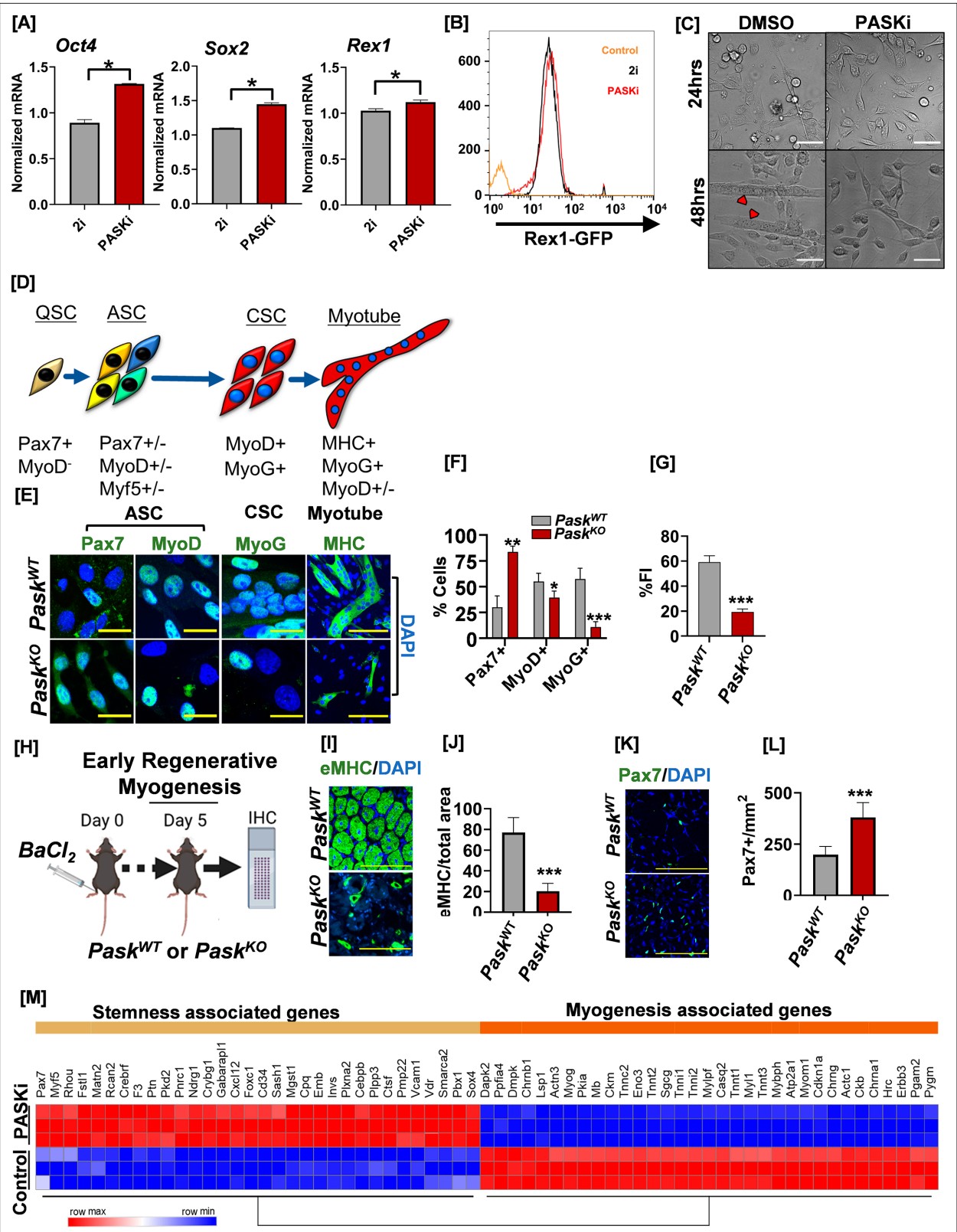

**Figure 1.** Inhibition of PAS domain-containing kinase (PASK) preserves pluripotency, decreases muscle stem cell (MuSC) heterogeneity, and inhibits precocious differentiation. (**A**) RT-qPCR analysis of indicated transcript levels from 2i+LIF cultured mouse embryonic stem cells (mESCs) after transitioning into 2i+LIF or PASKi+LIF conditions and cultured for 4 days. *p<0.05, error bars ± SD. (**B**) Rex1-GFP intensity levels from mESCs cultured in 2i vs. in PASKi as in (**A**). Rex1-GFP reporter expression was quantified by flow cytometry against a non-fluorescent control (control). (**C**) Isolated

*Figure 1 continued on next page*

*Figure 1 continued*

primary myoblasts were treated with DMSO or 50 μM PASKi for 4 days during the normal growth phase. Microscopy images were taken at 24 hr or 48 hr post-isolation and treatment. Scale bar = 40 μm. (**D**) Myogenic transcription factor progression of MuSCs after initial activation from quiescent (Pax7+) state. Activated MuSCs (ASC) are a highly heterogeneous population marked by varying levels and coexpression patterns of Pax7, MyoD, and Myf5. MyoD+ASC further differentiates into MyoG+ committed stem cells (CSC). CSC progenitors initiate a differentiation program to generate myotubes. (**E**) Expression pattern of myogenic regulatory factors in fluorescence-activated cell sorting (FACS)-sorted $Pask^{WT}$ and $Pask^{KO}$ myoblasts 48 hr after isolation. (**F**) Quantification of percent Pax7+, MyoD+, and MyoG+ cells from $Pask^{WT}$ and $Pask^{KO}$ animals. Error bars ± SD. *$p<0.05$, **$p<0.005$, ***$p<0.0005$ $(^{KO}$ vs. $^{WT})$. (**G**) Fusion index (% nuclei in myotubes/total nuclei) was calculated from MHC-stained cells isolated from $Pask^{WT}$ or $Pask^{KO}$ animals. Error bars ± SD. ***$p<0.0005$ $(^{KO}$ vs. $^{WT})$. (**H**) Experimental setup designed to compare the in vivo regeneration capabilities between $Pask^{WT}$ and $Pask^{KO}$ animals. (**I**) Embryonic myosin heavy chain (eMHC) staining of fresh-frozen muscle sections from mice of indicated genotype 5 days after muscle injury. DAPI marked nuclei, which are centrally localized in muscle sections from both animals, indicate of muscle regeneration in progress. (**J**) Quantification of eMHC+ myofiber numbers from experiment in Figure (**I**). Error bars ± SD. ***$p<0.0005$ $(^{KO}$ vs. $^{WT})$. (**K**) Muscle sections from $Pask^{WT}$ or $Pask^{KO}$ animals were stained with anti-Pax7 (green) antibodies 5 days post-injury and centrally located nuclei were visualized using DAPI. (**L**) Quantification of Pax7+ cell numbers in $Pask^{WT}$ vs. $Pask^{KO}$ muscle sections 5 days post-injury. Error bars ± SD, ***$p<0.0005$ $(^{KO}$ vs. $^{WT})$. (**M**) Heatmap of differentially expressed genes associated with stemness and myogenesis in C2C12 myoblasts treated with DMSO (control) or PASKi for 2 days.

The online version of this article includes the following source data and figure supplement(s) for figure 1:

**Source data 1.** Source data used to generate *Figure 1*.

**Source data 2.** Source data used to generate *Figure 1*.

**Figure supplement 1.** Maintenance of pluripotency and improvement of stemness properties by PAS domain-containing kinase (PASK) inhibition.

**Figure supplement 2.** PAS domain-containing kinase (PASK) inhibition sustains the proliferation of isolated primary myoblasts and blocks precocious differentiation.

**Figure supplement 3.** Characterization of primary myoblasts isolated from $Pask^{WT}$ or $Pask^{KO}$ animals.

**Figure supplement 4.** Analysis of RNAseq data from cultured myoblasts.

MyoG+ committed progenitor population by 3 days post-injury. These cells begin the regenerative myogenesis program, marked by the emergence of embryonic myosin (eMHC) expression. Loss of PASK severely affected the progression through the regenerative myogenesis program, as seen from decreased eMHC+ myofiber numbers in $Pask^{KO}$ animals compared with littermate controls 5 days post-injury (**Figure 1H–I**). Furthermore, loss of PASK resulted in a significant expansion of Pax7+ MuSC numbers in $Pask^{KO}$ muscles compared to the littermate control (**Figure 1K–L**). Combined with decreased MyoD+ myoblasts numbers and loss of MyoG expression in cells isolated from these animals in vitro (**Figure 1E–G**), our results indicate delayed generation of committed progenitor population in $Pask^{KO}$ animals. Thus, the loss of PASK impairs the transition from Pax7+ stem cells to the MyoD+/MyoG+ committed progenitor population required for the onset of the differentiation program.

The increase in Pax7+ cell numbers seen in isolated myoblasts in $Pask^{KO}$ or PASKi-treated cells could be attributed to the suppression of differentiation by the loss of active PASK. Thus, we turned to non-transformed, cultured myoblasts such as C2C12, which can be maintained in a proliferative state for an extended duration under appropriate culture conditions. Furthermore, C2C12 cells express low levels of Pax7 compared with isolated myoblasts and show increased heterogeneity in Pax7 expression (**Olguin and Olwin, 2004**). Thus, the C2C12 system has been employed to discover cell-intrinsic pathways regulating Pax7 expression and function (**Olguin and Olwin, 2004**; **McKinnell et al., 2008**; **Sincennes et al., 2021**). Using this system, we asked if PASKi treatment during the proliferative phase affects the expression of genes associated with stemness and self-renewal in cultured myoblasts. We performed RNAseq analysis of C2C12 myoblasts treated with PASKi during proliferative, early differentiating, and late differentiating conditions. Our global transcriptomic analysis revealed a strong enrichment of genes associated with stemness and self-renewal at all time-points when PASK is inhibited, starting with the proliferative condition (**Figure 1M**, **Figure 1—figure supplement 4**). Furthermore, PASK inhibition preserved self-renewal and stemness despite culture conditions that otherwise stimulate differentiation (**Figure 1M**, **Figure 1—figure supplement 4**). Our results in ESCs and MuSCs show that the PASK inhibition is a viable strategy to preserve in vitro self-renewal and pluripotency of ESCs and adult stem cells and is indicative of its functional role in balancing stemness and pluripotency with differentiation.

## PASK is dynamically redistributed from cytoplasmic granules to the nucleus in a cellular heterogeneity-dependent manner

To mechanistically understand how PASK inhibition sustains stem cell proliferation, we examined PASK subcellular distribution in proliferating versus early differentiating myoblasts. In proliferating C2C12 myoblasts (day 0), most of the PASK is localized in the cytosol and excluded from the nucleus (*Figure 2A*). Curiously, PASK appears to be localized into cytoplasmic granules in cultured myoblasts (*Figure 2A*, inset). Upon induction of the differentiation program, a significant loss of the cytoplasmic granular staining pattern occurs, and a large fraction of PASK was redistributed into the nucleus, as seen by overlapping intensities of the PASK signal with the nuclear marker DAPI and by biochemical fractionation (*Figure 2A–B*, *Figure 2—figure supplement 1*). We next asked if the subcellular distribution of PASK affects Pax7 expression in isolated primary myoblasts. Similar to cultured myoblasts, isolated primary myoblasts showed a strong granular staining pattern of PASK in proliferating myoblasts (*Figure 2C*). Furthermore, nearly all cells in which PASK was localized into the cytoplasmic granules were strongly positive for Pax7 expression (*Figure 2D*). In contrast, cells with any extent of nuclear-localized PASK lacked Pax7 expression, even under proliferating conditions (*Figure 2D*). In addition, in nascent myotubes (*Figure 2E*, marked by arrow), the PASK localization pattern is switched from within cytoplasmic granules to diffused nuclear, which correlated with the loss of Pax7 positivity (*Figure 2F*). These results suggest that nuclear translocation of PASK might be associated with heterogeneity in Pax7+ cell numbers seen in proliferating myoblasts (*Figure 2D*). Consistent with this, under proliferating conditions, Pax7hi (*Figure 2G*, yellow arrow indicates cells with stronger nuclear levels of Pax7) cells are more frequently associated with cytoplasmic PASK presence, and Pax7L0 (*Figure 2G*, white arrows indicate cells with relatively weaker nuclear levels of Pax7) or Pax7ab (cells lacking nuclear Pax7) are more frequently seen in cells with noticeable nuclear PASK presence (*Figure 2G–H*). Interestingly, we noticed asymmetric nuclear distribution of PASK in rapidly dividing myoblasts wherein a cell that asymmetrically retained nuclear PASK exhibited the loss of Pax7 expression post-mitosis, thereby creating heterogeneity in Pax7 expression in culture conditions (*Figure 2I–J*). Combined with data from *Figure 1*, our results indicated the mechanistic connection between PASK nuclear translocation and heterogeneity in Pax7 expression in proliferating myoblasts.

## Signal-regulated nuclear import-export machinery regulates nucleocytoplasmic shuttling of PASK

Since PASK inhibition or genetic loss resulted in increased Pax7+ myoblast numbers (*Figure 1E–F*, *Figure 1—figure supplement 3*), we hypothesize that nuclear translocation of PASK is the cause and not the effect of decreased Pax7 expression. To directly test this hypothesis, we asked whether forced nuclear retention of PASK could inhibit Pax7 expression and drive exit from self-renewal. To do that, we first fused a powerful SV40 Nuclear Localization Sequence (PKKKRKV, NLS) to GFP-tagged human PASK. To our surprise, NLS-tagging was insufficient to drive PASK into the nucleus (*Figure 3A*), indicating the possible presence of a powerful nuclear export sequence (NES) in PASK that ensures cytoplasmic localization of PASK in proliferating cells. Consistent with this, treatment of cells expressing NLS-hPASK but not WT-hPASK with nuclear exportin 1 (CRM1) inhibitor, leptomycin B (LMB), resulted in a modestly increased nuclear-localized PASK (*Figure 3A*).

Previous high-throughput studies have shown the interaction between human PASK and CRM1 (*Kırlı et al., 2015*). Thus, we considered the possibility that PASK is a nucleo-cytoplasmic shuttling protein containing one or more NES and that regulated import and/or export may result in its nuclear localization at the onset of differentiation. Proteins smaller than 60 kDa could migrate to the nucleus by diffusion (*Nigg, 1997*). Considering this size limitation, we performed a series of C-terminal truncations in PASK to identify the region that mediates PASK nuclear export (*Figure 3B*). Scoring for cells showing at least some nuclear GFP presence, we found that the GFP-WT-PASK (MW=~200 kDa) and GFP-1–737 (MW=~115 kDa) fragment remained predominantly cytoplasmic (*Figure 3C*). A smaller fragment, GFP-1–660 (MW=~100 kDa) showed nuclear GFP expression in ~22% of cells (*Figure 3C*), while fragment GFP-1–400 (MW=~70 kDa) showed increased nuclear localization of PASK with ~67% of cells showing at least some nuclear GFP presence (*Figure 3C*). These results suggested the presence of NES between amino acids 400 and 660 and between amino acids 660 and 737. We used multiple bioinformatic tools to identify L401–L409 (NES1) and L666–L671 (NES2) as putative NES in PASK (*Figure 3D*, *Figure 3—figure supplement 1*; *Figure 3—source data 1 and 2*; *Xu et al., 2015*;

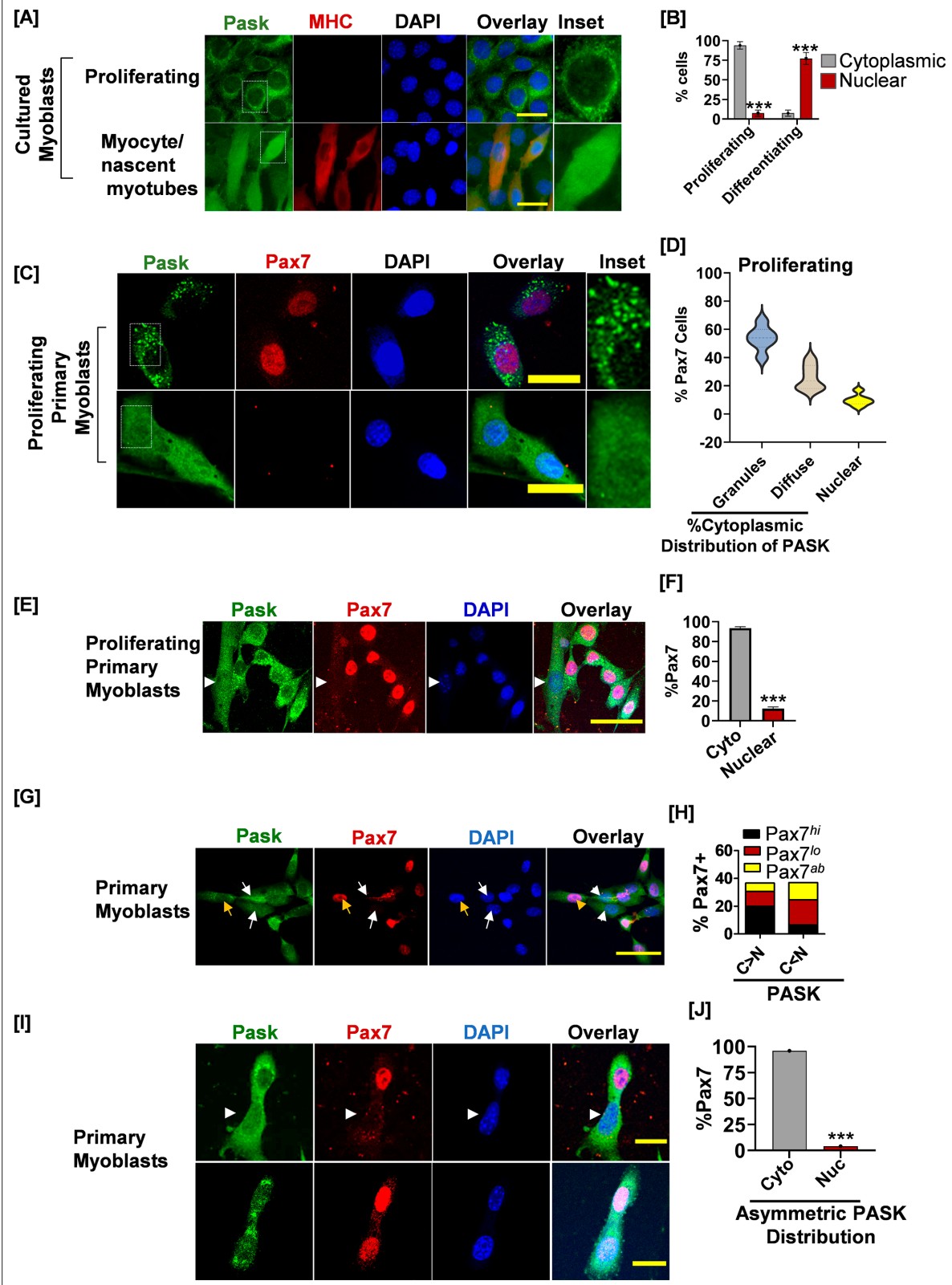

**Figure 2.** PAS domain-containing kinase (PASK) is localized in the cytoplasmic granules in self-renewing stem cells, which is redistributed to nucleus in Pax7-deficient cells. (**A**) Proliferating or differentiating C2C12 myoblasts were stained with anti-PASK (green), anti-MHC (MF20, red), or nuclei marker, DAPI antibody. Scale bar = 20 μm. The inset picture shows an enlarged view of the relative distribution of PASK in proliferating or differentiating myoblasts. Notice a granular or punctate staining pattern of endogenous PASK in C2C12 myoblasts in proliferative conditions. (**B**) Quantification

*Figure 2 continued on next page*

*Figure 2 continued*

of PASK subcellular distribution as a function of myoblasts state. Error bars ± SD, ***p<0.0005 *(Nuclear PASK in Differentiating vs. proliferating)*. (**C**) Fluorescence-activated cell sorting (FACS)-sorted muscle stem cells (MuSCs) from uninjured WT mice were stained for PASK and Pax7 24 hr after isolation. Notice a strong, granular staining pattern of PASK in the top panel, which is correlated with Pax7 positivity. The bottom panel shows uniform distribution of PASK, including in the nucleus. The inset picture shows the cytoplasmic granular staining pattern of PASK. (**D**) Violin plot showing a relationship between the subcellular localization of PASK and Pax7 expression. (**E**) FACS-sorted myoblasts were fixed 36 hr post-isolation and stained with PASK (green) and Pax7 (red). Arrow indicates nuclear PASK in early mononucleated myotubes. (**F**) Quantification of images from the experiment in (**F**). Error bars ± SD, ***p<0.0005 *(Pax7+ in nuclear vs. cytosolic PASK)*. (**G**) Distribution of Pax7 expression (red) (high, low, or absent, as indicated by Pax7$^{hi}$, Pax7$^{lo}$, Pax7$^{ab}$) in primary myoblasts with more cytosolic (C>N) vs. more nuclear (C<N) PASK (green). The yellow arrow indicates an example of a Pax7$^{hi}$ cell in which the PASK is cytoplasmic. The white arrow indicates Pax7$^{L0}$ cells in which a large proportion of PASK is diffused nuclear. (**H**) Quantification of % Pax7+ myoblasts numbers as a function of relative PASK subcellular distribution. (**I**) Asynchronously proliferating primary myoblasts were stained with PASK and Pax7 antibodies. Notice the exclusion of Pax7 from daughter mitotic cells with asymmetric nuclear localization of PASK (white arrow). Scale bar = 40 μm. Error bars ± SD, ***p<0.0005 *(Pax7+ in nuclear vs. cytosolic PASK)*.

The online version of this article includes the following source data and figure supplement(s) for figure 2:

**Figure supplement 1.** Subcellular fractionation of PAS domain-containing kinase (PASK) in proliferating vs. differentiating myoblasts.

**Figure supplement 1—source data 1.** Source data used to generate *Figure 2—figure supplement 1*.

---

*Xu et al., 2021*). The NES1 residues are similar to an experimentally verified NES in PGC1-α (*Chang et al., 2010*), and the NES2 residues are similar to the NES of PDK1 that we previously discovered (*Figure 3D*; *Lim et al., 2003*). We found that mutation of either NES1 or NES2 residues resulted in a modest but statistically significant increase in the number of cells with nuclear PASK localization compared to WT-PASK or NLS-PASK in the absence of LMB (*Figure 3E*). Combined mutation of NES1 and NES2 resulted in a significantly increased proportion of cells that contain nuclear PASK, with the extent of nuclear PASK in each cell similar to what we observed for PASK during myogenesis (*Figure 3E*). However, incomplete nuclear retention of NES1+NES2 mutated PASK prompted us to examine if the nuclear import of PASK might be rate-limiting, preventing stronger PASK nuclear accumulation despite NES1 and NES2 mutations (see *Figure 3—source data 3* for a list of mutant nomenclature in *Figure 3—figure supplement 1*). Consistent with that, while mutations of individual NES1 or NES2 in NLS-PASK improved nuclear retention of NLS-PASK (*Figure 3F*), mutating both NES1 and NES2 together in NLS-PASK resulted in robust nuclear retention of NLS-PASK in nearly 100% of cells (*Figure 3F*). These results conclusively show that NES1 and NES2 are functional for exporting PASK from the nucleus, and their loss interferes with the nucleo-cytoplasmic shuttling of PASK.

Since the nuclear accumulation of PASK was induced by differentiation signaling cues (*Figure 2A*), we asked if signaling pathways could target the nuclear export machinery (NES) to retain PASK in the nucleus. To test this, we used the WT version of NLS-hPASK to circumvent any regulatory import mechanisms that may be present (used in *Figure 3A*). We serum-starved HEK293T cells expressing NLS-PASK and treated them with LMB in the presence or absence of acute serum stimulation for 2 hr. Remarkably, NLS-PASK was excluded from the nucleus in serum-starved cells despite LMB treatment (*Figure 3G*). In contrast, acute serum stimulation increased nuclear accumulation of NLS-PASK even without LMB (*Figure 3G*). Furthermore, serum stimulation improved the effectiveness of LMB treatment, as seen from stronger nuclear retention of PASK (in nearly 100% of cells), similar to what was observed with NES1+NES2 mutated NLS-PASK. The improved effectiveness of LMB in blocking PASK nuclear export suggests that serum stimulation weakens the effectiveness of the NES1 and/or NES2 sequences to reduce the nuclear export rate of PASK.

## Glutamine-dependent acetylation of PASK stimulates its nuclear accumulation

We next asked if serum stimulation can trigger the nuclear translocation of endogenous and overexpressed GFP-tagged WT-PASK. As shown in *Figure 4A*, we observed significant nuclear accumulation of GFP-hPASK by 2 hr of serum stimulation. In addition, endogenous mouse PASK was also translocated to the nucleus in response to acute serum stimulation in C2C12 myoblasts (*Figure 4—figure supplement 1A*). We have previously shown that PASK is phosphorylated and activated by mTORC1 in response to serum stimulation (*Kikani et al., 2019*). This phosphorylation activates PASK to stimulate the myogenesis program. Two of the mTOR phosphorylation sites on PASK, Ser$^{640}$ and Thr$^{642}$, are juxtaposed to NES2 (L666–L671) residues. Therefore, we tested if mTOR signaling mediates the

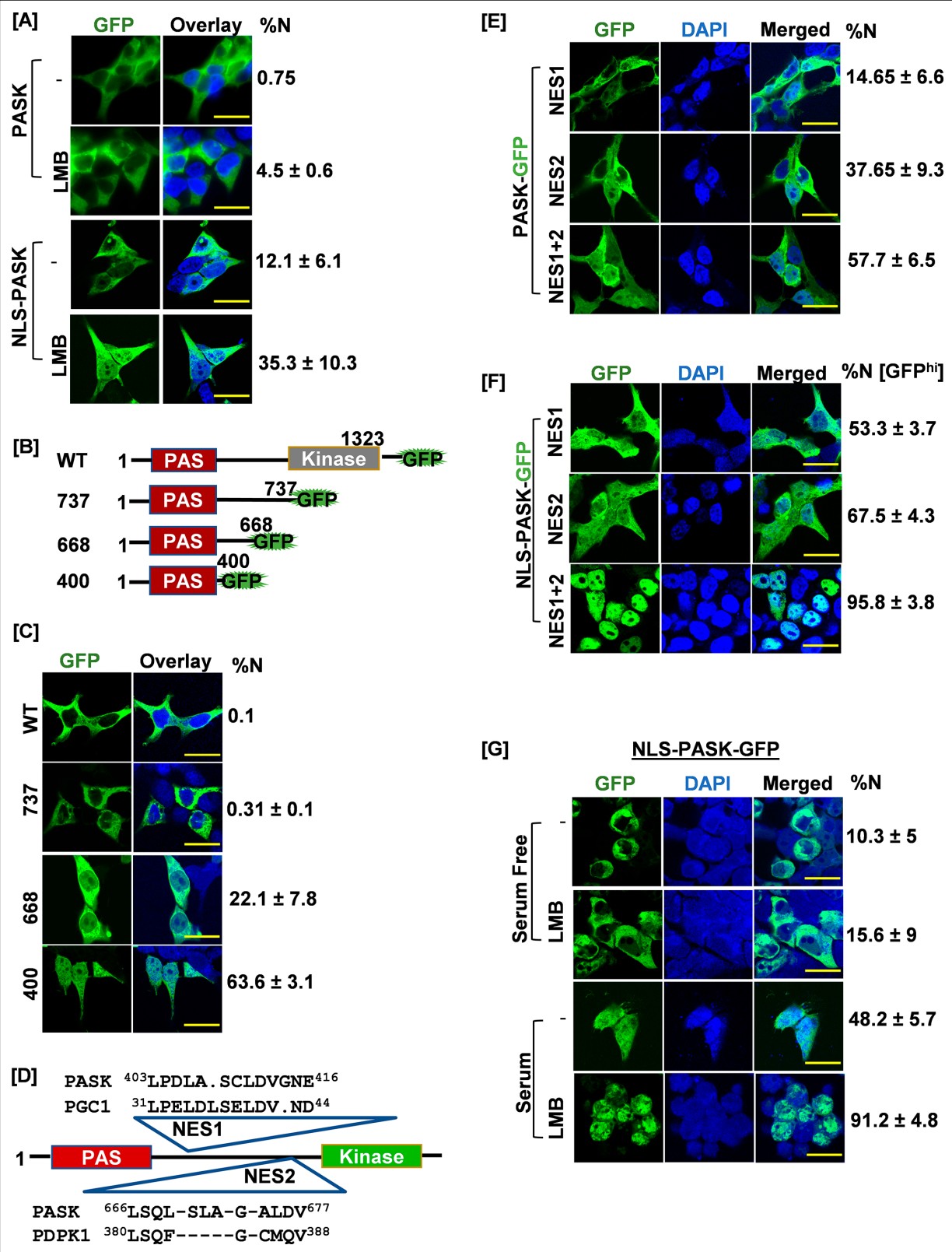

**Figure 3.** Signal regulated nuclear import mechanism for PAS domain-containing kinase (PASK). (**A**) HEK-293T cells were transfected with GFP-tagged full-length human PASK (aa 1–1323, WT) or GFP-tagged full-length SV40 NLS-PASK (aa 1–1323, NLS-WT). Cells were treated with 25 nM leptomycin B (LMB) for 2 hr as indicated and analyzed by confocal microscopy. The percentage of cells containing any GFP signal in the nucleus in each condition was quantified (% cells showing nuclear [N± SD] GFP). All scale bars = 40 μm. (**B**) Domain illustration depicting GFP-tagged WT hPASK (aa 1–1323)

*Figure 3 continued on next page*

*Figure 3 continued*

and its truncated versions, GFP-tagged fragment 737 (aa 1–737), 668 (aa 1–668), and 400 (aa 1–400). The PAS domain is highlighted in red. (**C**) HEK-293T cells were transfected with GFP-tagged full-length PASK (aa 1–1323, WT) and various truncations. Cells were analyzed by confocal microscopy. The percentage of cells containing any GFP signal in the nucleus was included in the quantification (%N± SD). (**D**) Diagram showing the locations of two nuclear export sequences (NES1 and NES2) and their sequence homology relative to PGC1 (NES1) and PDPK1 (NES2), respectively. (**E**) HEK-293T cells were transfected with GFP-tagged full-length WT PASK (aa 1–1323, WT) or NES1 ($L^{403}AL^{405}S$) or NES2 ($L^{666}SL^{671}A$). Cells were analyzed by confocal microscopy. The percentage of cells containing any GFP signal in the nucleus in each condition was quantified (% N± SD). (**F**) HEK-293T cells were transfected with GFP-tagged SV40 NLS-PASK (NLS-hPASK-GFP) containing mutated nuclear export sequences (NES1, NES2, or combined NES 1+2). Cells were analyzed by microscopy. The percentage of cells containing strong GFP signal ($GFP^{hi}$) in the nucleus in each condition was quantified (% N± SD). (**G**) HEK-293T cells were transfected with GFP-tagged full-length SV40 NLS-hPASK (aa 1–1323, NLS-PASK-GFP). Cells were serum-starved (0.1% serum) for 12 hr and then subsequently stimulated with either 0.1% serum (serum free) or 20% serum (serum) for 2 hr. Cells were treated with 25 nM leptomycin B (LMB), where indicated and analyzed by confocal microscopy. The percentage of cells containing any GFP signal in the nucleus in each condition was quantified (% N± SD).

The online version of this article includes the following source data and figure supplement(s) for figure 3:

**Source data 1.** Source data used to generate *Figure 3*.

**Source data 2.** Source data used to generate *Figure 3*.

**Source data 3.** Source data used to generate *Figure 3*.

**Figure supplement 1.** Prediction of PAS domain-containing kinase (PASK) nuclear export sequences.

nuclear translocation of WT-PASK in response to serum stimulation. However, inhibiting mTORC1 by rapamycin did not consistently or significantly block serum-induced nuclear translocation of WT-h-PASK. This suggests that mTORC1 alone is insufficient for driving the nuclear translocation of PASK in response to serum stimulation (*Figure 4A*) and that a separate signaling input controls its nuclear translocation.

Along with phosphorylation, acetylation of non-histone proteins plays an important role in controlling nucleo-cytoplasmic shuttling and cell fate decisions in stem cells (*Choudhary et al., 2014*). To determine if endogenous PASK is acetylated, we treated C2C12 myoblasts with trichostatin A, a broad-spectrum histone deacetylase inhibitor. Probing the purified acetylome with an anti-PASK antibody revealed strong enrichment of acetylated endogenous mouse PASK (*Figure 4—figure supplement 1B*). We also noticed a dose (*Figure 4—figure supplement 1C*) and time-dependent (*Figure 4B*) increase in PASK acetylation caused by serum stimulation in HEK293T cells.

To examine the physiological relevance of PASK acetylation in vivo, we injured mouse tibialis anterior (TA) muscles and analyzed PASK acetylation on day 3 and day 5 post-injury. We noticed that a significant fraction of mouse PASK is acetylated during regeneration on days 3 and 5 post-injury (*Figure 4—figure supplement 1D*). Together, these results show that PASK is a novel non-histone protein target of signal-dependent acetylation in cells and tissues.

Due to the overlapping temporal kinetics of PASK acetylation (*Figure 4B*) and nuclear translocation (*Figures 3G and 4A*), we asked whether PASK acetylation drives its nuclear translocation. To test this, we first sought to identify upstream acetyltransferase for PASK. Kat2a and EP300 are two major histone acetyltransferases upregulated in MuSCs during regeneration and are known to induce the acetylation of non-histone proteins during myogenesis (*Das et al., 2017*; *Puri et al., 1997a*; *Puri et al., 1997b*; *Sartorelli et al., 1999*). siRNA-mediated knockdown of Kat2a did not affect serum-induced acetylation of PASK (*Figure 4—figure supplement 1E*). In contrast, the pretreatment of cells with a highly selective CBP/P300 inhibitor, A-485 (*Lasko et al., 2017*), nearly completely blocked serum-induced PASK acetylation (*Figure 4C*) in HEK293T cells. Furthermore, the knockdown of EP300 in C2C12 cells completely inhibited endogenous PASK acetylation induced by serum (*Figure 4—figure supplement 1F*). Finally, serum-induced PASK nuclear localization (*Figure 4D*) was blocked by the pretreatment of cells with A-485. Thus, PASK is a novel acetylation target of CBP/EP300 that drives its nuclear translocation in response to serum stimulation.

To connect CBP/EP300-driven PASK acetylation to upstream signaling pathways, we measured serum-induced PASK acetylation in the presence of various signaling kinase inhibitors. However, serum-induced PASK acetylation was independent of inputs from signaling kinases, as inhibition of PI-3K, mTOR, or Akt did not prevent serum-induced PASK acetylation (*Figure 4—figure supplement 1G*). Therefore, we explored the involvement of metabolic pathways in driving PASK acetylation. Acetyl-CoA, a substrate for acetyltransferases, is generated from glucose (via mitochondrial pyruvate

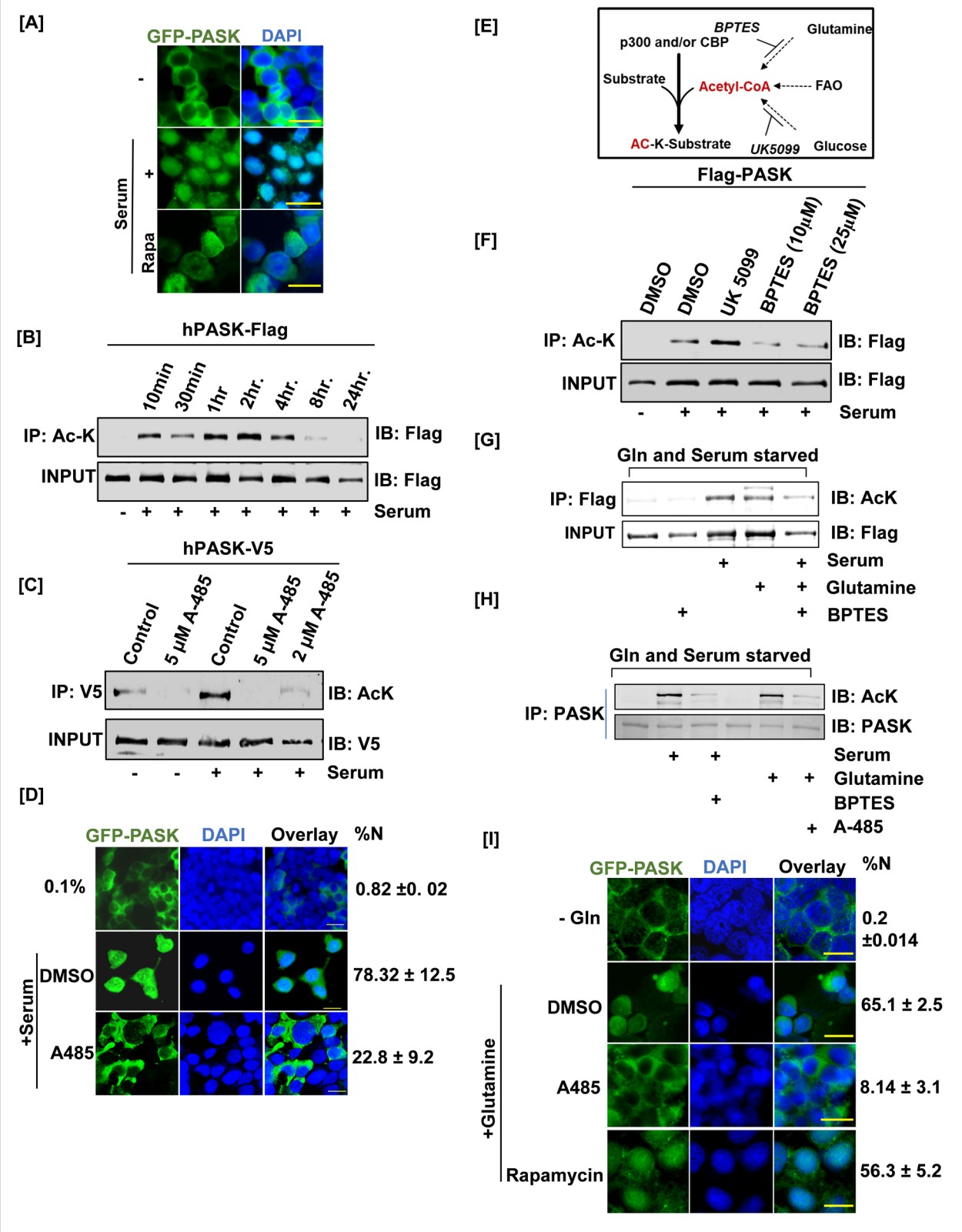

**Figure 4.** Mitochondrial glutamine metabolism stimulates CBP/EP300-dependent acetylation of PAS domain-containing kinase (PASK) to drive its nuclear translocation. (**A**) HEK-293T cells stably expressing GFP-tagged full-length WT PASK (aa 1–1323) were serum-starved (0.1% serum) for 12 hr and then stimulated for 2 hr with 20% serum. For the rapamycin-treated condition, cells were pretreated with 100 nM rapamycin for 2 hr in serum-free conditions prior to 20% serum induction. Scale bar = 40 µm. Since LMB alone is ineffective at driving PASK nuclear retention (**Figure 3A**), we performed

*Figure 4 continued on next page*

*Figure 4 continued*

all treatments in the presence of low-dose (5 nM) LMB. (**B**) HEK-293T cells stably expressing FLAG-tagged full-length PASK (aa 1–1323) were serum-starved (0.1% serum) for 12 hr. Cells were subsequently stimulated with either 0.1% serum or 20% serum for the time intervals indicated. Total cellular acetylome was purified from cells using anti-Ac-K antibody and the presence of acetylated PASK was detected by anti-Flag antibody. (**C**) V5-tagged full-length PASK (aa 1–1323) was transiently expressed in HEK-293T cells. 24 hr after transfections, cells were serum-starved in 0.1% serum for 12 hr. Cells were pretreated with 5 µM or 2 µM A-485 to inhibit CBP/EP300 acetyltransferases 1 hr prior to serum stimulation. Cells were stimulated with either 0.1% serum or 20% serum for 2 hr. V5-tagged PASK was immunoprecipitated from cells, and the presence of acetylated PASK was detected using an anti-Ac-K antibody. (**D**) HEK-293T cells stably expressing GFP-tagged full-length PASK (aa 1–1323) were serum-starved (0.1% serum) for 12 hr. Cells were pretreated with 2 µM A-485 to inhibit CBP/EP300 acetyltransferases 1 hr prior to serum stimulation. Cells were stimulated with either 0.1% serum (-) or 20% serum (serum) for 2 hr, and the extent of nuclear GFP-tagged PASK (%Nuc ± SD) was analyzed by immunofluorescence. Scale bar = 20 µm. (**E**) Illustration of metabolic pathways involved in acetyl-CoA production that can fuel cellular acetylation. (**F**) HEK-293T cells stably expressing FLAG-tagged full-length PASK (aa 1–1323) were treated with 10 µM UK 5099 (mitochondrial pyruvate carrier inhibitor) or 10 µM BPTES (glutaminolysis inhibitor) overnight along with 0.1% serum starvation where indicated. Cells were stimulated with 20% serum in the presence of DMSO or indicated inhibitors. hPASK was purified using anti-Flag antibody and probed with anti-AcK antibody. (**G**) C2C12 cells stably expressing FLAG-tagged full-length PASK (aa 1–1323) were serum-starved in glutamine-free media for 12 hr. 10 µM BPTES was added along with serum/glutamine starvation where indicated and maintained during 2 hr serum stimulation. For glutamine stimulation, media containing 2 mM glutamine was added with or without 10 µM BPTES. hPASK was purified using anti-Flag antibody and probed with anti-AcK antibody. (**H**) Primary myoblasts were serum- and glutamine-starved for 12 hr in the presence of 10 µM BPTES where indicated. Cells were stimulated with 20% FBS or 2 mM glutamine in the presence of either 2 µM A-485 (for glutamine-stimulated cells) or 10 µM BPTES (for serum-stimulated cells) for 2 hr. The lysate was immunoprecipitated for endogenous PASK prior to immunoblotting for total acetyl-lysine (Ac-K). (**I**) HEK-293T cells stably expressing GFP-tagged full-length PASK (aa 1–1323) were serum- and glutamine-starved for 12 hr. Cells were pretreated with DMSO (as control), 100 nM rapamycin, or 2 µM A-485 for 2 hr prior to stimulation with 2 mM glutamine. Cells were fixed and the extent of nuclear GFP-tagged PASK was quantified by confocal microscopy (%N± SD). Scale bar = 40 µm. Since LMB alone is ineffective at driving PASK nuclear retention (*Figure 3A*), we performed all treatments in the presence of low-dose (5 nM) LMB.

The online version of this article includes the following source data and figure supplement(s) for figure 4:

**Source data 1.** Source data used to generate *Figure 4*.

**Source data 2.** Source data used to generate *Figure 4*.

**Source data 3.** Source data used to generate *Figure 4*.

**Source data 4.** Source data used to generate *Figure 4*.

**Source data 5.** Source data used to generate *Figure 4*.

**Source data 6.** Source data used to generate *Figure 4*.

**Figure supplement 1.** PAS domain-containing kinase (PASK) is acetylated in vitro and in vivo during muscle regeneration.

**Figure supplement 1—source data 1.** Source data used to generate *Figure 4—figure supplement 1*.

**Figure supplement 1—source data 2.** Source data used to generate *Figure 4—figure supplement 1*.

**Figure supplement 1—source data 3.** Source data used to generate *Figure 4—figure supplement 1*.

**Figure supplement 1—source data 4.** Source data used to generate *Figure 4—figure supplement 1*.

**Figure supplement 1—source data 5.** Source data used to generate *Figure 4—figure supplement 1*.

oxidation) and glutamine (via glutaminolysis). Strikingly, inhibition of the mitochondrial pyruvate complex (MPC1) by UK5099 stimulated PASK acetylation more than serum alone, perhaps due to a compensatory increase in glutaminolysis (*Figure 4F*; *Yang et al., 2014*). Consistent with this, inhibition of mitochondrial glutaminolysis by a GLS1-specific inhibitor, BPTES, resulted in a near-complete loss of serum-induced PASK acetylation (*Figure 4F*). Furthermore, only glutamine added back to cells cultured in glutamine- and serum-free media conditions was sufficient to stimulate PASK acetylation as much as serum induction (*Figure 4G*), which was further inhibited by BPTES (*Figure 4G*). Finally, glutamine metabolism drives PASK acetylation via CBP/P300 since pretreatment of serum-stimulated cells with BPTES or glutamine-stimulated cells with the A-485 markedly blunted the PASK acetylation in isolated primary MuSCs (*Figure 4H*).

Next, we asked if glutamine metabolism performs a signaling function to propel PASK nuclear translocation. As shown in *Figure 4I*, glutamine starvation resulted in the nuclear exclusion of PASK. On the other hand, glutamine addition alone in serum- and glutamine-free media stimulated robust nuclear translocation of PASK, which was blocked by CBP/EP300 inhibition (A-485), but not by mTORC1 inhibitor rapamycin (*Figure 4I*). Thus, our results show that glutamine signaling to CBP/EP300, but not to mTOR, regulates PASK nuclear translocation via acetylation.

# Selective glutamine withdrawal triggers the preservation of stemness in MuSCs

The critical role of glutamine in stimulating PASK nuclear translocation prompted us to examine the role of glutamine metabolism at various stages of self-renewal and differentiation. First, leveraging the heterogeneity of cultured myoblasts in terms of Pax7 expression (*Olguin and Olwin, 2004*), we asked if glutamine withdrawal affects myoblast proliferation and self-renewal. Consistent with previous reports (*Shang et al., 2020*; *Ahsan et al., 2020*), glutamine withdrawal significantly reduced the proliferation rate of cultured myoblasts (*Figure 5A*). However, the glutamine-withdrawn cells were greatly enlarged and remained alive in cell culture for at least 4 days (*Figure 5A–B*). Furthermore, cells grown in glutamine-replete media exhibited a lower proportion of Pax7+ myoblasts, consistent with a previous report (*Olguin and Olwin, 2004*). Interestingly, acute glutamine withdrawal resulted in a significantly reduced heterogeneity in Pax7 expression due to a significant increase in the percentage of Pax7+ myoblasts (*Figure 5B–C*). To test if glutamine depletion plays a similar role in isolated primary myoblasts, we subjected FACS-sorted primary myoblasts to glutamine depletion after 24 hr of culture in standard myoblast proliferation media, allowing them to grow for an additional 48 hr. Consistent with results from cultured myoblasts, isolated primary myoblasts cells grown in glutamine-replete media exhibited a decreased proportion of Pax7+ myoblasts. In contrast, glutamine-depleted cells showed a marked increase in the percent Pax7+ myoblasts, resulting in reduced myoblast heterogeneity in terms of Pax7 expression (*Figure 5D–E*). These results suggest a connection between glutamine metabolism and MuSCs heterogeneity via regulation of Pax7 expression dynamics, warranting further mechanistic investigation.

Since progression through the cell cycle is linked with the emergence of cellular heterogeneity in isolated stem cell cultures (*Rocheteau et al., 2012*; *Perera et al., 2022*), our data suggest that stem cells activate the stemness preservation program when the proliferation of stem cells is metabolically unfavorable (*Figure 5A and D*). To evaluate the connection between Pax7+ myoblast heterogeneity and cell proliferation with glutamine signaling in a cell-autonomous setting, we determined the extent of Pax7 nuclear staining in BrdU+ and BrdU- populations of isolated primary myoblasts under normal or glutamine-depleted culture conditions. BrdU+ cells (proliferating cells) in glutamine-rich media were smaller in size and displayed heterogeneity in the level of nuclear Pax7 expression (*Figure 5F–G*). We also noticed a significant Pax7$^{L0}$ and Pax7$^{ab}$ population among the BrdU+ cycling stem cells grown in glutamine-replete media (*Figure 5F–G*). In contrast, glutamine withdrawal blunted BrdU incorporation yet resulted in a robust increase in the proportion of Pax7$^{hi}$ cells due to increased Pax7 staining intensity within individual cells (*Figure 5F–H*). Since the ratio of Pax7+/BrdU+ nuclei is increased by glutamine withdrawal (*Figure 5H*), our results indicate that glutamine withdrawal results in a reactivation of Pax7 expression in Pax7$^{lo}$ MuSCs in the absence of cell proliferation. Consistent with this, *Pax7* mRNA levels were significantly increased when glutamine was withdrawn, whereas mRNA levels for proliferation marker *MKi67* were nearly completely blunted (*Figure 5—figure supplement 1A*). In addition to *Pax7*, transcript levels of uncommitted myoblast markers, such as *Cd34*, *Myf5*, and *Myod1*, were also significantly elevated upon glutamine deprivation (*Figure 5—figure supplement 1A*). We also noticed a significant increase in the expression of *Foxo3*, a marker for MuSC quiescence (*Figure 5—figure supplement 1A*), perhaps reflective of reduced proliferation due to glutamine withdrawal. These results suggested that glutamine withdrawal enforces the stemness preservation program and induces quiescence in vitro in a cell-autonomous manner. Finally, glutamine withdrawal also resulted in a near-complete loss of *Myog*, indicating a dependence on glutamine metabolism to generate the committed progenitor population (*Figure 5—figure supplement 1A*). We validated these functions of glutamine in cultured myoblasts as glutamine withdrawal or inhibition of Gls1 by BPTES treatment increased Pax7 mRNA levels in C2C12 myoblasts (*Figure 5—figure supplement 1B*). While glutamine depletion blocked myogenesis, inhibition of mitochondrial pyruvate import by UK5099 strongly stimulated differentiation as measured by *Myog*, *Mylpf*, and *Acta1* transcript levels (*Figure 5—figure supplement 1B*) and immunostaining (*Figure 5—figure supplement 1C*), perhaps via increased glutaminolysis (*Yang et al., 2014*).

During the early muscle regeneration response, glutamine, enriched in the regenerating niche, is taken up by activated MuSCs, which drives their proliferation and is required for myogenic commitment and the regenerative myogenesis program (*Shang et al., 2020*). According to a recent report, a quiescent, self-renewing population is established after this early wave of the myogenesis program has

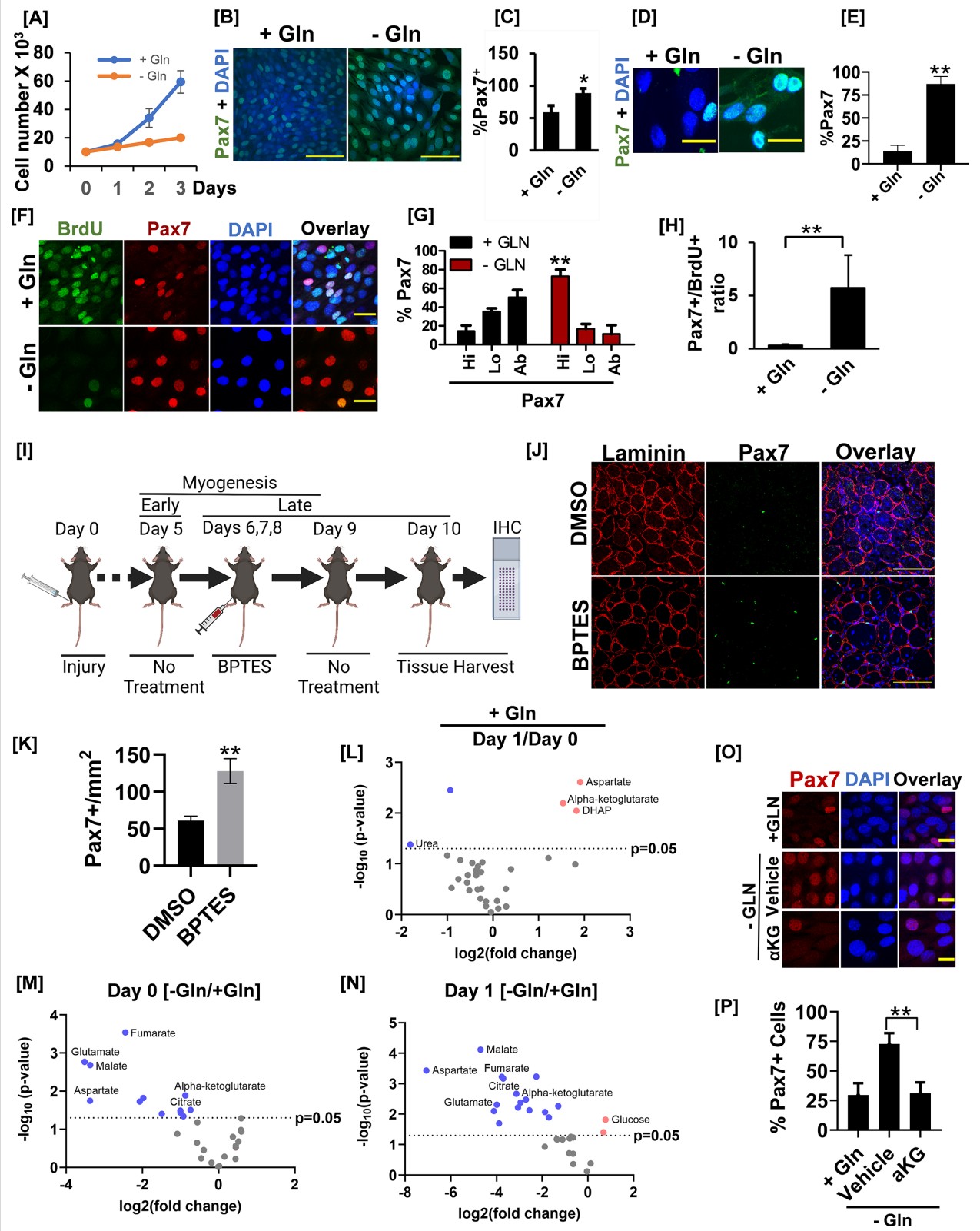

**Figure 5.** Glutamine metabolism drives muscle stem cell heterogeneity by countering Pax7 expression. (**A**) The cell proliferation rate of C2C12 myoblasts cultured in the presence (+Gln) or absence (-Gln) of glutamine over 3 days. $10^4$ cells were seeded in 60 mm tissue culture dishes. Twenty-four hr after seeding, cells were transferred to DMEM without glutamine media containing 10% serum with (+Gln) or without (-Gln) 2 mM glutamine added. Cells were counted every day for 3 days using an automated cell counter. (**B**) C2C12 myoblasts were allowed to proliferate in the presence

*Figure 5 continued on next page*

*Figure 5 continued*

(+Gln) or absence (-Gln) of glutamine for 2 days as in (**A**). Cells were fixed and stained with anti-Pax7 antibody and nuclei were visualized by DAPI. Scale bar = 100 µm. (**C**) Quantification of Pax7+ cell numbers in the presence (+Gln) or absence of glutamine (-Gln). * Error bars ± SD. p<0.05, n=5 (% Pax7 cells in -Gln vs. +Gln). (**D**) Primary myoblasts were plated in media containing glutamine for 24 hr. Growth media was replaced with media containing (+Gln) or lacking (-Gln) for additional 24 hr as in (**A**). Cells were fixed and immunostained with anti-Pax7 antibody. Scale bar = 40 µm. (**E**) Quantification of %Pax7+ cell numbers between +Gln and -Gln conditions. Error bars ± SD. **p<0.005, n=3 (% Pax7 cells in - Gln vs. +Gln). (**F**) 24hr after isolation and culture in media containing glutamine, primary myoblasts were allowed to proliferate in the presence (+Gln) or absence (-Gln) of glutamine for additional 2 days. Cells were pulsed with 10 µM BrdU in +Gln or -Gln media for 24 hr before cell fixation. Cells were fixed, and BrdU incorporation and Pax7+ cell numbers were analyzed by immunofluorescence microscopy using antibodies against BrdU and Pax7. Scale bar = 40 µm. (**G**) % Pax7+ cells showing Pax7 expression at high (hi), low (lo), or absent (ab) levels in the presence (+Gln) or absence (-Gln) of glutamine. Error bars ± SD. **p<0.005 (% Pax7$^{hi}$ cells in -Gln vs. +Gln). (**H**) Quantification of the ratio of % Pax7+ myoblasts over % BrdU+ myoblasts from immunofluorescence analysis in (**F**). Error bars ± SD. **p<0.005 (% Pax7$^{hi}$ cells in -Gln vs. +Gln). (**I**) Experimental setup to test the effect of acute Gls1 inhibition by BPTES injection on Pax7 levels. Twelve-week-old C57BL/6 mice were injured with 25 µl of 1.2 % wt/vol BaCl$_2$ to the tibialis anterior (TA) muscles. On days 6, 7, and 8 post-injury, 10 µl of phosphate buffer saline (PBS) containing 0.01% DMSO or 10 µM BPTES was injected intramuscularly into the belly of TA muscles. Tissues were harvested on day 10. Tissue sections were stained for laminin (red) or Pax7 (green) and analyzed by confocal microscopy. N=5 mice per treatment group. Both legs were injured and treated with DMSO or BPTES and utilized for analysis as technical replicates. (**J**) Immunofluorescence images of muscle sections stained with Pax7 and laminin from the experiment in (**G**). Scale bar = 100 µm. (**K**) Quantification of average Pax7+ cell number/field from images in (**H**). **p<0.005 (BPTES vs. DMSO). (**L**) Volcano plot of metabolite abundances in differentiating (day 1) versus proliferating (day 0) C2C12 myoblasts in the glutamine replete condition. (**M**) Volcano plot of metabolite abundances in proliferating (day 0) C2C12 myoblasts cultured in the presence (+Gln) or absence (-Gln) of glutamine. (**N**) Volcano plot of metabolite abundances in differentiating (day 1) C2C12 myoblasts cultured in the presence (+Gln) or absence (-Gln) of glutamine. (**O**) C2C12 myoblasts were grown in the presence (+Gln) or absence of glutamine (-Gln) for 24 hr. One mM cell-permeable dimethyl-alpha ketoglutarate (aKG) or vehicle control was added to glutamine-depleted cells for 24 hr. After treatment, cells were fixed and stained with an anti-Pax7 antibody. (**P**) Quantification of Pax7+ cell number from the experiment in (**M**). Scale bar = 20 µm. **Error bars ± SD. p<0.005.

The online version of this article includes the following source data and figure supplement(s) for figure 5:

**Figure supplement 1.** Glutamine depletion preserves stem cell identity while antagonizing differentiation.

**Figure supplement 1—source data 1.** Source data used to generate *Figure 5—figure supplement 1*.

**Figure supplement 1—source data 2.** Source data used to generate *Figure 5—figure supplement 1*.

largely concluded (days 5–7 post-injury) (*Cutler et al., 2021*). Our results show that acute glutamine withdrawal resulted in the reactivation of Pax7 expression, decreased proliferation, and increased expression of the quiescence marker, *Foxo3*, in isolated MuSCs. Therefore, we asked if the withdrawal of glutamine signaling due to niche remodeling could contribute to the reactivation of Pax7 transcription and expand the self-renewing stem cell population after the early wave of the myogenesis program is completed. To test this, we injured the TA muscles from WT mice and allowed them to undergo regenerative myogenesis up to day 5 (*Figure 5I*). This ensures normal progression through the early myogenesis program. On day 5, we intramuscularly injected 10 µM BPTES or DMSO every day for 3 days to directly introduce Gls1 inhibitor into the regenerating niche, followed by 1 day of no additional treatment to allow for clonal expansion of cell populations. We then harvested TA muscles and quantified Pax7+ stem cell numbers (*Figure 5J–K*). After early myogenesis, 3 days of BPTES treatment did not appear to affect regenerative myofiber sizes (laminin A staining, *Figure 5J*). In contrast, we saw a significant increase in Pax7+ cell numbers in BPTES-treated compared with DMSO-treated animals (*Figure 5J–K*).

We next sought to identify the metabolic intermediates downstream of glutamine metabolism that regulates Pax7 levels in myoblasts. We first performed unbiased whole-cell metabolomics of proliferative vs. differentiating myoblasts. Compared with proliferating cells, early differentiating myoblasts show significantly increased intracellular glutamine levels and their associated downstream metabolites, glutamate, alpha-ketoglutarate, and aspartate (*Figure 5L*). Under glutamine-depleted conditions where proliferation is inhibited but stemness is enhanced (*Figure 5F–H*), we noticed a significant reduction in alpha-ketoglutarate, citrate, and malate levels, indicating a possible reliance on reductive carboxylation from alpha-ketoglutarate in proliferative myoblasts (*Figure 5L–N*, *Figure 5—figure supplement 1D–F*). Consistent with this, the addition of cell-permeable alpha-ketoglutarate reversed the increased Pax7+ cell numbers in glutamine-depleted culture conditions (*Figure 5O–P*). Taken together, our results suggest that glutamine metabolism links proliferation with the passage from Pax7$^{hi}$ stem cells to Pax7$^{L0/Ab}$ progenitors, progressively establishing a commitment to differentiate (MyoG+).

# Glutamine disrupts the WDR5-APC/C interaction during mitosis to drive exit from self-renewal via PASK nuclear translocation

As a downstream target of glutamine metabolism, we sought to determine if PASK mediates the function of glutamine metabolism in regulating Pax7 expression. Unlike glutamine withdrawal, PASK inhibition did not affect cell proliferation rate or size (*Figure 6—figure supplement 1A*) in isolated MuSCs. Nevertheless, similar to glutamine withdrawal, PASK inhibition significantly increased Pax7+ myoblast numbers and reduced the heterogeneity of Pax7 nuclear-staining intensities across the field in isolated MuSCs (*Figure 6A–B*). These results suggest that PASK plays a specific role downstream of glutamine metabolism in regulating the self-renewal program but not cell proliferation machinery. To test that, we performed a BrdU incorporation assay in PASKi-treated isolated primary myoblasts. Interestingly, while PASKi did not affect BrdU incorporation, PASKi treatment resulted in increased Pax7+ MuSCs numbers (*Figure 6—figure supplement 1B–C*). Thus, PASK selectively regulates the stemness downstream of glutamine metabolism without affecting the proliferation rate.

We have noticed a marked expansion in Pax7+ cell numbers in *Pask*$^{KO}$ mice as early as day 5 post-injury (*Figures 1I–J and 6C–D*). After the initial wave of myogenesis is completed, the self-renewing population is reestablished, and the Pax7+ MuSCs number begins to return to its pre-injury levels. Interestingly, as with acute Gls1 inhibition (*Figure 5J–K*), *Pask*$^{KO}$ mice showed elevated Pax7+ MuSCs numbers even 14 days after injury, whereas Pax7+ MuSCs numbers decline in *Pask*$^{WT}$ muscles during that time (*Figure 6C–D*). Furthermore, *Pask*$^{KO}$ animals showed smaller myofiber fiber cross-sectional areas at 14 dpi compared with *Pask*$^{WT}$ muscle sections (*Figure 6E*). Thus, genetic loss or inhibition of PASK results in an increased proportion of Pax7+ myoblast numbers in vitro, expansion of Pax7+ MuSC numbers during muscle regeneration, and a compromised muscle regeneration response.

The requirement of alpha-ketoglutarate as a co-factor in DNA and histone demethylation reactions is associated with its function in regulating cell fate in ESCs (*Ryall et al., 2015*; *TeSlaa et al., 2016*). In our experimental model, the addition of cell-permeable alpha-ketoglutarate reversed the increase in Pax7+ stem cell numbers caused by glutamine depletion (*Figure 5O–P*). To determine if this function of alpha-ketoglutarate is dependent on PASK activity and separate from its epigenetic functions, we measured the effectiveness of alpha-ketoglutarate in reversing Pax7+ myoblast numbers in the presence of PASKi. Interestingly, PASK inhibition rendered alpha-ketoglutarate ineffective in reducing Pax7+ stem cell numbers in glutamine-depleted media conditions (*Figure 6F–G*). These results suggest that the glutamine-dependent nuclear translocation of PASK is an additional pathway that drives myoblast exit from self-renewal. In support of this hypothesis, we show that glutamine depletion results in increased cytoplasmic retention of PASK in isolated myoblasts, with a concomitant increase in Pax7+ myoblasts (*Figure 6H–I*). Thus, we propose that glutamine metabolism targets PASK subcellular distribution to generate a Pax7-negative population.

Consistent with this, an increase in Pax7+ cell numbers observed due to PASK silencing in cultured myoblasts could be reversed by the expression of either WT or nuclear hPASK (nPASK [NLS-PASK-NES1+2-GFP], *Figure 6J–K*), but not by GFP control (*Figure 6J*, *Figure 1—figure supplement 1D-E*). In contrast, only nuclear PASK effectively reversed the increased Pax7+ myoblasts numbers due to glutamine depletion (*Figure 6L–M*). Taken together, we conclude that PASK is a downstream target of glutamine metabolism in activated myoblasts that counters Pax7 expression.

We have previously reported an increased association between PASK and WDR5, a member of the histone H3 lysine 4 methyltransferase complex at the onset of differentiation (*Kikani et al., 2016*; *Karakkat et al., 2019*). Furthermore, WDR5 interacts with Pax7 and epigenetically maintains myoblast self-renewal (*McKinnell et al., 2008*). Therefore, we asked if the PASK-WDR5 association contributes to the downregulation of Pax7 expression downstream of glutamine metabolism. To further expand the biochemical understanding of the PASK-WDR5 association, we searched for a canonical WDR5 interaction motif (WIN) sequence within PASK exhibiting a high degree of sequence similarity with other WIN motif-containing proteins (*Figure 7A*). The multiple alignments identified R942 as a conserved residue within the WIN motif that interacts directly with WDR5 (*Figure 7A*). Mutation of Arg$^{942}$Gly (R942G) within the WIN motif ablated the interaction between hPASK and Flag-tagged mouse WDR5 (*Figure 7B*). We have previously reported that mTOR phosphorylates PASK at multiple sites to promote the PASK-WDR5 association (*Kikani et al., 2019*). Three sites (Ser949, Ser953, Ser956) juxtapose with the WIN motif. Phosphomimetic mutation of a single mTOR phosphorylation site, Ser949 to glutamic acid (E), resulted in a stronger association between PASK and WDR5

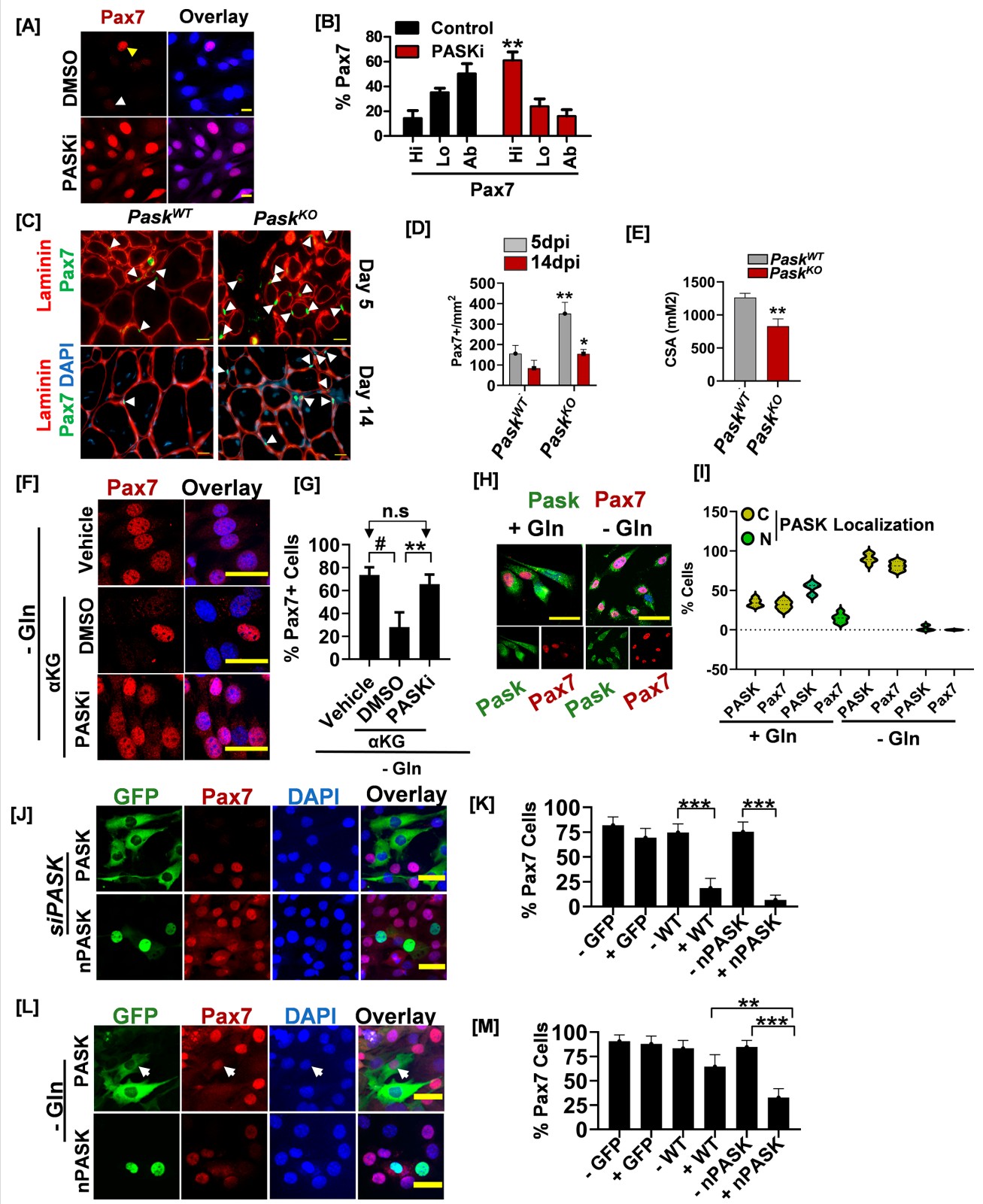

**Figure 6.** Glutamine-induced stem cell heterogeneity is mediated by nuclear PAS domain-containing kinase (PASK) function. (**A**) Isolated primary myoblasts were grown in the presence or absence of 50 µM PASKi for 48 hr. Cells were fixed and Pax7+ cells were visualized by immunostaining. (**B**) Quantification of cells showing Pax7 expression at high (hi), low (lo), or absent (ab) levels from DMSO or 50 µM PASKi-treated cells. (**C**) Tibialis anterior (TA) muscles of littermates of *Pask^WT* and *Pask^KO* animals were injured using 1.2% BaCl₂. Frozen tissue sections were stained using anti-Pax7 and

*Figure 6 continued on next page*

*Figure 6 continued*

anti-laminin antibodies. Arrows indicate examples of Pax7+ cells identified and used for quantification in (**D**). Scale bar = 40 µm. (**D**) Quantification of Pax7+ muscle stem cells (MuSCs) numbers at day 5 or day 14 in *Pask$^{WT}$* vs. *Pask$^{KO}$* mice. Scale bars = 20 µm. Error bars ± SD. **p<0.005 (*5 dpi in Pask$^{KO}$ vs Pask$^{WT}$*), *p<0.005 (*14 dpi in Pask$^{KO}$ vs Pask$^{WT}$*). (**E**) Quantification of myofiber cross-sectional area measured from laminin staining **p<0.005 (*5 dpi in Pask$^{KO}$ vs. Pask$^{WT}$*). (**F**) C2C12 myoblasts were grown in complete media containing 2 mM glutamine for 24 hr. Culture media was replaced with glutamine depleted (-Gln) or -Gln media containing 1 mM dimethyl alpha-ketoglutarate (aKG) in the presence of DMSO or 50 µM PASKi for additional 24 hr. After treatments, cells were fixed and stained with anti-Pax7 antibody. Scale bar = 40 µM. (**G**) Quantification of % Pax7+ C2C12 from the experiment in (**I**). Error bars ± SD. p<0.05 (*# Vehicle vs. DMSO, **PASKi vs. DMSO, n.s. [not significant] between PASKi and vehicle*). (**H**) Magnetic-activated cell sorting (MACS) double-sorted (see Methods) primary myoblasts were cultured and treated as described in *Figure 5D*. Cells were fixed and stained using anti-PASK (green), anti-Pax7 (red), and DAPI (nuclei). Scale bar = 40 µM. Individual channels representing PASK and Pax7 are shown below a three-channel overlay for each condition (+Gln and -Gln). (**I**) Violin plot representing the relationship between PASK subcellular distribution and Pax7 expression in the presence or absence of glutamine. Note near-complete lack of nuclear PASK expressing cell population in glutamine-withdrawn conditions. (**J**) Mouse PASK was silenced using multiplex siRNAs in C2C12 cells. Twenty-four hr after siRNA transfection, GFP-tagged WT or NLS-hPASK$^{NES1+2}$-hPASK (nuclear PASK, nPASK) was retrovirally expressed. Forty-eight hr after viral infection, cells were fixed, and Pax7+ cells were analyzed by immunofluorescence against Pax7. PASK-expressing cells were identified by GFP expression. Scale bar = 40 µm. (**K**) Quantification of %Pax7$^{hi}$ cell numbers in individual cells expressing GFP control (GFP+), WT-PASK (WT+), or nuclear PASK (nPASK+) compared with uninfected cells (GFP-, WT-, or nPASK-) from the experiment in (**E**). Mean± SD. ***p<0.0005. (**L**) Pax7 expression was assessed by immunofluorescence microscopy from C2C12 myoblasts cultured for 48 hr in glutamine-depleted conditions expressing GFP control, GFP tagged WT, or nPASK plasmids. Scale bar = 40 µm. (**M**) Quantification of %Pax7$^{hi}$ cell numbers in individual cells expressing GFP control (GFP+), WT-PASK (WT+), or nuclear PASK (nPASK+) compared with uninfected cells (GFP-, WT-, or nPASK-) from the experiment in (**G**). Data represented in quantification are mean± SD. ***p<0.0005.

The online version of this article includes the following source data and figure supplement(s) for figure 6:

**Figure supplement 1.** Glutamine-driven generation of differentiation-competent progenitors requires nuclear PAS domain-containing kinase (PASK) function.

**Figure supplement 1—source data 1.** Source data used to generate *Figure 6—figure supplement 1*.

(*Figure 7B*), further confirming that nutrient signaling modulates the PASK-WDR5 interaction (*Kikani et al., 2019*).

WDR5-WIN motif-protein interactions have emerged as a key regulator of many aspects of cellular response and behavior (*Guarnaccia et al., 2021*). In stem cells, WDR5 is reported to interact with the cell cycle-linked E3 ubiquitin ligase, APC/C via a WIN motif. This interaction is strengthened during mitosis and is required for the rapid transcriptional reactivation of self-renewal and lineage commitment genes following mitosis in ESCs (*Oh et al., 2020*). Disruption of the WDR5-APC/C interaction reduces stem cell pluripotency. However, the upstream signal that regulates the WDR5-APC/C interaction remains unknown.

We have noticed rapid nuclear entry of human PASK during mitosis, prior to nuclear membrane breakdown (*Figure 7—figure supplement 1A*). Thus, as an upstream regulator of WDR5, we asked if PASK disrupts the WDR5-APC/C interaction during mitosis to suppress Pax7 expression. Since APC2 was demonstrated to be the only APC/C complex member that binds directly with WDR5 when the proteins are purified in vitro (*Oh et al., 2020*), we focused our co-immunoprecipitation (co-IP) experiments on APC2 and CDC20, two members of the endogenous APC/C complex that interact with WDR5 during mitosis. The presence of WT-PASK significantly disrupted WDR5-APC/C, as indicated by reduced co-precipitation of WDR5 with APC2 and CDC20 compared with control cells (*Figure 7C*). PASK$^{WIN}$ mutant, on the other hand, significantly strengthened the WDR5-APC/C interaction, suggesting that the interaction of PASK with WDR5 disrupts the WDR5-APC/C interaction (*Figure 7C*). Since the WDR5-APC/C interaction is specifically strengthened during mitosis, we studied the role of various PASK mutants, including a catalytically dead version (KD, K1028R), in regulating the WDR5-APC/C interaction during mitosis. As shown in *Figure 7D*, co-expression of WT-PASK with WDR5 significantly reduced the WDR5-APC/C interaction. However, co-expression of either kinase-dead (K1028R) or WIN mutant (R942G) PASK resulted in strengthened WDR5-APC/C interaction when compared to the expression of WT PASK (*Figure 7D*). Expression of PASK$^{S949E}$, on the other hand, further reduced the WDR5-APC/C interaction compared with WT-PASK, consistent with the stronger PASK-WDR5 association and increased PASK activity resulting from phosphorylation

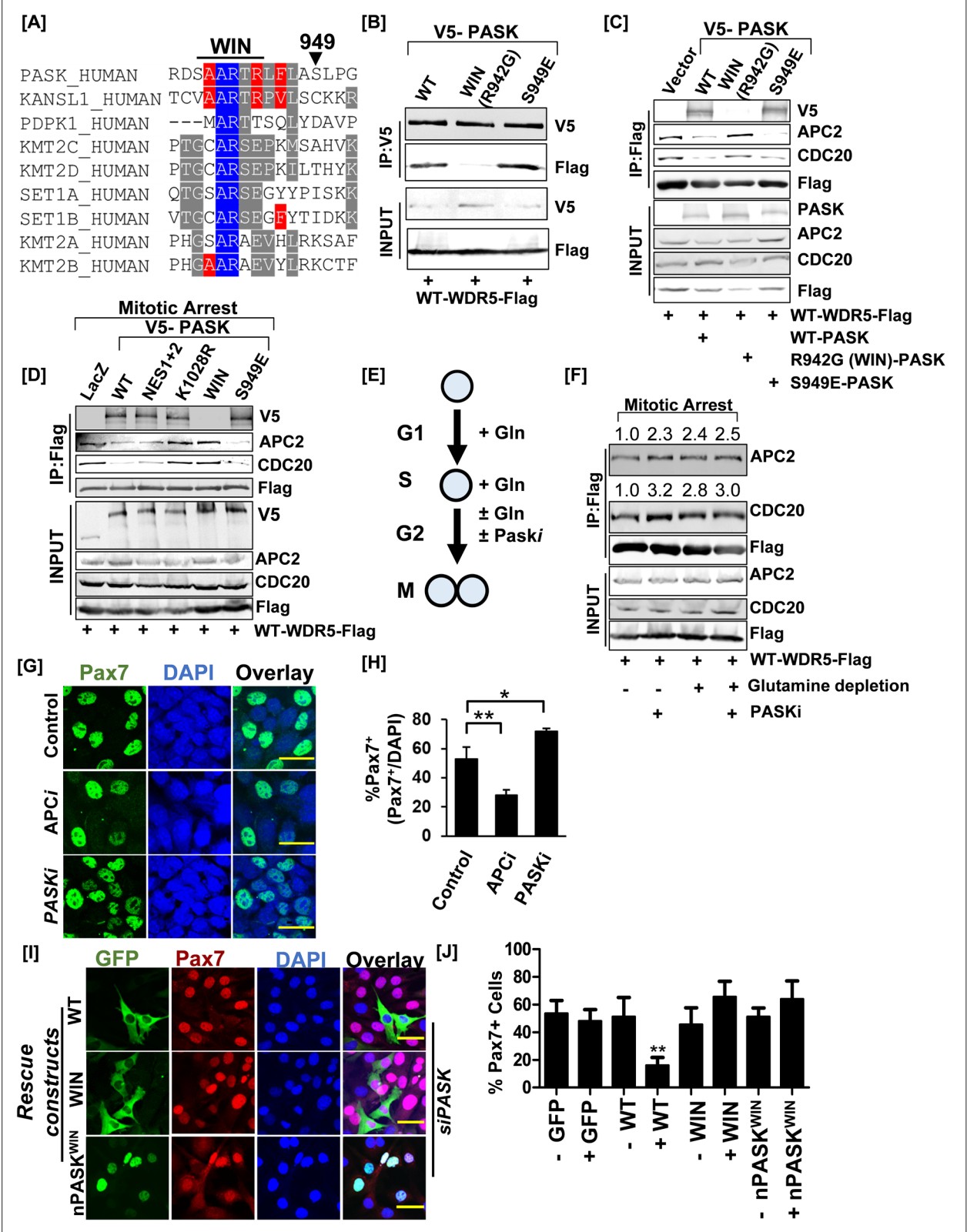

**Figure 7.** Nuclear PAS domain-containing kinase (PASK) disrupts the WDR5-anaphase-promoting complex/cyclosome (APC/C) interaction causing exit from self-renewal. (**A**) Alignment of the hPASK WIN motif with previously established WIN motifs. S949 represents a previously reported mTOR phosphorylation site on PASK. (**B**) Biochemical characterization of PASK-WDR5 interaction via R942 (WIN motif). HEK-293T cells were co-transfected with Flag-WDR5 and full-length V5-tagged WT-PASK (WT), R942G (WIN), or a phosphomimetic mutant of S949, S949E (S949E). Cells were lysed in

*Figure 7 continued on next page*

*Figure 7 continued*

native lysis buffer, and Flag-WDR5 was immunoprecipitated from cell extract. Co-precipitated PASK versions were detected by probing with anti-V5 antibodies. (**C**) PASK disrupts the WDR5-APC/C interaction. HEK-293T cells were transfected with full-length V5-tagged WT (WT), R942G (WIN), or S949E PASK, along with FLAG-WDR5. Asynchronous cells were lysed in native lysis buffer, and WDR5 complexes were purified using anti-Flag beads. The immunoprecipitants were probed with PASK, APC2, CDC20, and FLAG antibodies. (**D**) PASK catalytic activity, phosphorylation of PASK by mTORC1, and WIN motif interaction converge to disrupt the WDR5-APC/C interaction during mitosis. HEK-293T cells expressing indicated plasmids were synchronized to the G2/M boundary by nocodazole treatment as described in the Methods. G2/M arrested cells were collected via centrifugation, and cell pellets were lysed in native lysis buffer. Flag-WDR5 containing protein complexes were purified using anti-Flag beads. Relative enrichment of APC2, CDC20, and PASK was determined by western blotting. (**E**) Graphical representation of experimental scheme testing effect of glutamine withdrawal on WDR5-APC/C interaction. Cells were arrested at the G1/S boundary in the presence of glutamine. G1/S arrested cells were released into media containing 2 µM nocodazole in the presence or absence of glutamine with or without 50 µM PASKi. (**F**) Western blot analysis of WDR5-APC/C interaction in G2/M arrested cells in (**E**). The numbers indicate the normalized intensities of western blot signal relative to Flag signal from immunoprecipitation (IP). (**G**) Primary myoblasts were treated with DMSO (control), 50 µM proTAME (APC/C inhibitor, APCi), or 50 µM PASKi for 30 hr. Pax7+ myoblasts numbers were quantified by immunofluorescence microscopy using anti-Pax7 antibody. Scale bar = 40 µm. (**H**) Quantification of % Pax7+ cells from the experiment in (**G**). (**I**) PASK-WDR5 interaction regulates myoblast identity. GFP-hPASK, GFP-hPASK[R942G](WIN), or nPASK-WIN [R942G](nPASK-WIN) were expressed in PASK-silenced C2C12 cells. Forty-eight hr after retroviral infection, cells were fixed, and the proportion of Pax7[hi] cells was quantified by immunofluorescence microscopy using an anti-Pax7 antibody. Scale bar = 40 µm. (**J**) Quantification of % Pax7[hi] cells from the experiment in (**I**). Data represented in quantification are mean± SD. **p<0.005.

The online version of this article includes the following source data and figure supplement(s) for figure 7:

**Source data 1.** Source data used to generate *Figure 7*.

**Source data 2.** Source data used to generate *Figure 7*.

**Source data 3.** Source data used to generate *Figure 7*.

**Source data 4.** Source data used to generate *Figure 7*.

**Figure supplement 1.** Negative regulation of the WDR5-anaphase-promoting complex/cyclosome (APC/C) interaction by PAS domain-containing kinase (PASK) drives myogenesis.

**Figure supplement 1—source data 1.** Source data used to generate *Figure 7—figure supplement 1*.

**Figure supplement 1—source data 2.** Source data used to generate *Figure 7—figure supplement 1*.

**Figure supplement 2.** Model.

at the Ser949 site (*Figure 7D*). We previously showed that PASK phosphorylated WDR5 at S49 and that the phosphomimetic mutant of WDR5, S49E, dramatically reduced Pax7 expression and induced precocious myogenesis under proliferating conditions (*Kikani et al., 2016*). Interestingly, we find that WDR5[S49E] completely disrupted the WDR5-APC/C interaction during mitosis, suggesting that the catalytic function of PASK regulates the WDR5-APC/C interaction (*Figure 7—figure supplement 1C*).

Glutamine depletion suppressed proliferation while stimulating Pax7 expression and the preservation of stemness. That raises the possibility that glutamine metabolism stimulated nuclear PASK associates with WDR5 to downregulate Pax7 transcription and orchestrate the exit from self-renewal. Consistent with this, acute glutamine stimulation strengthened the PASK-WDR5 interaction in proliferating myoblasts (*Figure 7—figure supplement 1B*). To test directly if glutamine metabolism, functioning via PASK, modulates the WDR5-APC/C interaction, we allowed cells to be synchronized to the G1-S boundary in media containing glutamine since the WDR5-APC/C interaction is specifically strengthened during mitosis. We then released and arrested cells at the G2/M boundary in media containing or lacking glutamine supplementation in the presence or absence of PASKi (*Figure 7E*). This allowed for G2/M phase-specific analysis of the role of glutamine metabolism in modulating the WDR5-APC/C interaction in stem cells. As shown in *Figure 7F*, PASKi treatment strengthened the WDR5-APC/C interaction in glutamine-containing media during mitosis. Interestingly, glutamine withdrawal during mitosis was sufficient to strengthen the WDR5-APC/C interaction, and the addition of PASKi in the glutamine-depleted condition did not have further additive effects (*Figure 7F*). This result suggests that the WDR5-APC/C interaction is a target of glutamine metabolism during mitosis in stem cells and that PASK relays the glutamine signaling to WDR5-APC/C.

Cells that asymmetrically retain PASK in the nucleus exhibit loss of Pax7 expression in proliferating myoblasts (*Figure 2I*). Heterogeneity in the nuclear expression of PASK underlies heterogeneity in Pax7 expression (*Figure 2I–J*), which can be reversed by catalytic inhibition of PASK (*Figure 6A–B*). Furthermore, inhibition of glutamine metabolism reduces proliferation, inhibits PASK nuclear translocation, strengthens WDR5-APC/C interaction, and reduces heterogeneity in Pax7 expression. Therefore, we asked if APC/C activity underlies the re-establishment of Pax7 transcription after mitosis in isolated primary myoblasts. Strikingly, APC/C inhibition using the specific inhibitor proTAME resulted in a significant reduction in the number of Pax7+ cells after at least one round of the cell cycle (30 hr of treatment) compared with control cells (*Figure 7G–H*). Within this timeframe, PASKi treatment significantly increased Pax7+ cell number (*Figure 7G–H*). Additionally, WT-PASK effectively downregulated Pax7 expression (*Figure 7I–J*) while mutation of WIN (R942G), which abolishes the PASK-WDR5 interaction and strengthened WDR5-APC/C interaction, resulted in a near-complete failure in suppressing Pax7 expression with both WT and nuclear versions of PASK (*Figure 7I*). Finally, acute inhibition of APC/C resulted in the transcriptional induction of the committed progenitor and differentiation markers, *Myog* and *Mylpf*, respectively (*Figure 7—figure supplement 1D*). In contrast, inhibition of WDR5 or PASK, alone or in combination with APC/C inhibitor, prevented the establishment of the committed progenitor population and blocked differentiation (*Figure 7G–H*, *Figure 7—figure supplement 1D*). These results suggest multifaceted functions of PASK and WDR5 that act in conjunction with APC/C to regulate the generation of committed progenitors and indicate that proper temporal transition of cell identity is required for successful terminal differentiation. Therefore, we provide a novel signaling role of PASK in connecting glutamine metabolism with the WDR5-APC/C complex interaction to control stem cell fate. Our results suggest that the proliferative functions of glutamine metabolism are co-opted by stem cells to establish stem cell fate via PASK in response to the changing metabolic landscape.

## Discussion

Through comprehensive studies in this manuscript, we identify PASK as a novel sensory kinase that integrates hormonal (via mTOR) and metabolic signals (via glutamine) into the post-mitotic machinery to counter the reestablishment of stem cell identity. Our data support a model where glutamine stimulates the proliferation of activated myoblasts to bring activated stem cells into mitosis. Furthermore, glutamine induces concomitant acetylation of PASK to drive its nuclear translocation and guide the exit from self-renewal, establishing the committed progenitor pool during early muscle regeneration (*Figure 7—figure supplement 2*). Thus, as a metabolic sensory kinase enriched in stem cells, PASK functions to integrate metabolic (via glutamine) and nutrient signaling (via mTOR) inputs to orchestrate exit from self-renewal and promote the onset of the terminal differentiation program.

### Role of glutamine metabolism in PASK nuclear localization

Our data suggest that PASK subcellular localization in stem cells must be carefully regulated to prevent unintended differentiation. In agreement, our results indicate that nuclear accumulation of PASK is signal-regulated. Serum-induced PASK nuclear localization was independent of mTOR but dependent upon CBP/P300-induced acetylation. Interestingly, CBP/P300 has been reported to other acetylate non-histone proteins, mainly transcription regulatory factors, during myogenesis (*Puri et al., 1997a*; *Puri et al., 1997b*). Our results suggest that PASK is a novel kinase target of CBP/P300 in MuSCs that drives differentiation by coordinating exit from self-renewal.

The cellular fuel source that produces acetyl-coA connects the nutrient environment with stem cell fate. Our data conclusively shows that mitochondrial glutamine metabolism is the primary source of acetyl-CoA for driving both PASK acetylation and nuclear translocation. Furthermore, glutamine is a major fuel early during the myogenesis program, which coincides with the temporal point at which PASK expression is at its highest in regenerating muscles (*Kikani et al., 2016*; *Shang et al.,*

*2020*). Thus, we propose that increased glutamine influx from the regenerating niche induces PASK acetylation and nuclear translocation to promote the onset of the myogenesis program early during regeneration.

## Control of self-renewal by glutamine and PASK

Acute loss of glutamine arrested cell proliferation but augmented the transcription *Pax7* and increased the percent Pax7+ SC numbers (*Figure 5B–D*, *Figure 5—figure supplement 1A*). Similarly, inhibition of PASK with PASKi increased Pax7$^{hi}$ cell numbers (*Figure 6A–B*), and *Pask$^{KO}$* animals exhibited increased Pax7+ cell numbers during regeneration (*Figures 1K–L and 6C–D*). These data suggest that glutamine and PASK act on the same downstream mechanisms to regulate self-renewal.

To understand how PASK inhibition promotes stem cell self-renewal, we focused on the interactome of its downstream target, WDR5, and its involvement in stem cell fate. Our previous work found that phosphorylation of WDR5 by PASK at Ser49 is sufficient for driving myogenesis (*Kikani et al., 2019*; *Kikani et al., 2016*). Furthermore, WDR5 is required to maintain self-renewal in ESCs and adult MuSCs via its association with Oct4 and Pax7 transcription factors, respectively (*McKinnell et al., 2008*; *Ang et al., 2011*). WDR5 also serves as a substrate adaptor by associating with APC/C during mitosis to ubiquitylate histone H2B via K11/K48-branched chains to maintain the open chromatin architecture necessary for the rapid transcription of self-renewal and pluripotency-associated genes following mitosis (*Oh et al., 2020*). Our results suggest that, in response to glutamine signaling, PASK disrupts the WDR5-APC/C interaction during mitosis, which is required for downregulating Pax7 expression and coordinating exit from self-renewal.

Therefore, our studies provide a new model for connecting the role of the metabolic environment with the intracellular, mechanistic control of stem cell fate. Our parallel results in ES cells further suggest the generalizability of PASK function to stem cells of varied tissue origins since the downstream target of PASK function, the WDR5-APC/C interaction, is a conserved feature among both mESCs and MuSCs. Transient inhibition of PASK in both mESCs and MuSCs increases the differentiation-capable progenitor population, providing a potential rationale for PASKi to be used for future therapeutic applications for the maintenance of stem cell function.

## Methods

### Key resources table

| Reagent type (species) or resource | Designation | Source or reference | Identifiers | Additional information |
|---|---|---|---|---|
| Antibody | Mouse monoclonal Anti-Pax7 | DSHB, University of Iowa | DSHB Cat# PAX7, RRID:AB_2299243 | IF: 1:100 IHC: 1:50 |
| Antibody | Mouse monoclonal Anti-MyoG | DSHB, University of Iowa | DSHB Cat# f5d, RRID:AB_2146602 | IF: 1:100 |
| Antibody | Mouse monoclonal Anti-MHC | DSHB, University of Iowa | DSHB Cat# MF 20, RRID:AB_214778 | IF: 1:100 |
| Antibody | Rabbit polyclonal Anti-Laminin | Sigma-Aldrich | Sigma-Aldrich Cat# L9393, RRID:AB_477163 | IF: 1:100 |
| Antibody | Mouse monoclonal Anti-FLAG | Sigma-Aldrich | Sigma-Aldrich Cat# F3165, RRID:AB_259529 | IF: 1:100 |
| Antibody | Mouse monoclonal Anti-V5 | Thermo Fisher | Thermo Fisher Scientific Cat# R960-25, RRID:AB_2556564 | IF: 1:100 |
| Antibody | Rabbit polyclonal Anti-PASK | Cell Signaling Technology | Cell Signaling Technology Cat# 3086, RRID:AB_2159082 | IF: 1:100 WB: 1:1000 |
| Antibody | Rabbit monoclonal Anti-Acetylated lysine | Cell Signaling Technology | Cell Signaling Technology Cat# 9814, RRID:AB_10544700 | IP: 5 µg WB: 1:1000 |

*Continued on next page*

*Continued*

| Reagent type (species) or resource | Designation | Source or reference | Identifiers | Additional information |
|---|---|---|---|---|
| Antibody | Rabbit polyclonal Anti-APC2 | Cell Signaling Technology | Cell Signaling Technology Cat# 12301, RRID:AB_2754994 | WB: 1:1000 |
| Antibody | Rabbit monoclonal Anti-CDC20 | Cell Signaling Technology | Cell Signaling Technology Cat# 14866, RRID:AB_2715567 | WB: 1:1000 |
| Antibody | Rabbit polyclonal α/β Tubulin | Cell Signaling Technology | Cell Signaling Technology Cat# 2148, RRID:AB_2288042 | WB: 1:1000 |
| Antibody | Rabbit monoclonal Anti-KAT2A (GCN5L2) | Cell Signaling Technology | Cell Signaling Technology Cat# 3305, RRID:AB_2128281 | WB: 1:1000 |
| Antibody | Rabbit monoclonal Anti-p300 | Cell Signaling Technology | Cell Signaling Technology Cat# 70088, RRID:AB_2799773 | WB: 1:1000 |
| Antibody | Rabbit polyclonal Anti-BrdU | Thermo Fisher Scientific | Thermo Fisher Scientific Cat# PA5-32256, RRID:AB_2549729 | IF: 1:100 |
| Antibody | Rabbit monoclonal Anti-WDR5 | Cell Signaling Technology | Cell Signaling Technology Cat# 13105, RRID:AB_2620133 | IP: 5 µg WB: 1:1000 |
| Commercial assay kit | Tyramide Signal Amplication kit | Biotium | 33000 | |
| Commercial assay kit | TruSeq Stranded Total RNA Library Prep Kit with Ribo-Zero Gold | Illumina | RS-122–2302 | |
| Commercial assay kit | QIAGEN's RNeasy | QIAGEN | Cat# 75162 | |
| Cell line (Human) | HEK 293T | ATCC | ATCC Cat# CRL-3216 | |
| Cell line (Mouse) | C2C12 | ATCC | ATCC Cat# CRL-1772, RRID:CVCL_0188 | |
| Cell line (Mouse) | Mouse ES cells | ATCC | Isolated from mice | |
| Cell line (Mouse) | Primary mouse myoblasts | Isolated fresh from mice | Isolated from mice | |
| Chemical compounds | BPTES | Selleck Chem | S7753 | |
| Chemical compounds | A-485 | Selleck Chem | S8740 | |
| Chemical compounds | UK5099 | Selleck Chem | S5317 | |
| Chemical compound, drug | BioE-1197 | Jared Rutter, University of Utah | Not applicable | |
| Chemical compound, drug | Rapamycin | LC Labs | R-5000 | |
| Chemical compound, drug | Akt inhibitor IV | Sigma-Aldrich | 124011 | |
| Chemical compound, durg | LY294002 | Cell Signaling Technology | 9901 | |
| Chemical compound, drug | Leptomycin B | LC Labs | L-6100 | |
| Chemical compound, drug | proTAME | Cayman Chemicals | 25835 | |
| Chemical compound, drug | WDR5-0103 | Cayman Chemicals | 13945 | |
| Software, algorithm | GraphPad Prism v 9.0 | GraphPad | | |
| Software, algorithm | BioRender | BioRender | | |

*Continued on next page*

| qPCR primers | Sequence |
| --- | --- |
| *Myog*_FW | GAGACATCCCCCTATTTCTACCA |
| *Myog*_RV | GCTCAGTCCGCTCATAGCC |
| *Myf5*_FW | AAGGCTCCTGTATCCCCTCAC |
| *Myf5*_RV | TGACCTTCTTCAGGCGTCTAC |
| *Myod1*_FW | CCACTCCGGGACATAGACTTG |
| *Myod1*_RV | AAAAGCGCAGGTCTGGTGAG |
| *Acta1*_FW | CCCAAAGCTAACCGGGAGAAG |
| *Acta1*_RV | CCAGAATCCAACACGATGCC |
| *Mylpf*_FW | AAGGAGGCGTTCACTGTAATTG |
| *Mylpf*_RV | TAGCGTCGAGTTCCTCATTCT |
| *Pax7*_FW | TCTCCAAGATTCTGTGCCGAT |
| *Pax7*_RV | CGGGGTTCTCTCTCTTATACTCC |
| *Rn18s*_FW | GTA ACC CGT TGA ACC CCA TT |
| *Rn18s*_Rv | CCA TCC AAT CGG TAG TAG CG |
| *MKi67*_FW | CAA GGC GAG CCT CAA GAG ATA |
| *MKi67*_RV | TGT GCT GTT CTA CAT GCC CTG |
| *Foxo3*_FW | CAT GCG CGT TCA GAA TGA AGG |
| *Foxo3*_RV | GAC TGT CGT CTG CCG ACT C |

## Experimental model and subject details

### Mouse models

*Pask*$^{WT}$ and *Pask*$^{KO}$ animals (B6.129S6-PASKtm1Weng/J, Strain #:017709, RRID: IMSR_JAX:017709) animals were described previously (*Kikani et al., 2016*). Animal experiments were performed in accordance with protocols approved by the Institutional Animal Care and Use Committee at the University of Kentucky to CK (2019–3317). Animals were housed in the environment in compliance with the Guide for the Care and Use of Laboratory Animals in an AAALAC-accredited facility at the University of Kentucky on a 12:12 hr light: dark cycle.

### Cell culture

HEK 293T and C2C12 myoblasts were acquired from the American Type Culture Collection (ATCC). Early passage, mycoplasma-free cells were stored in a liquid-nitrogen vapor phase and used for all experiments. Cells were routinely tested for mycoplasma contamination using a PCR-based mycoplasma detection kit (Thermo Fisher Scientific, J66117-AMJ). HEK293T cells were authenticated by functional tests, proliferative performance, and epithelial morphological characteristics. C2C12 cells were routinely maintained at low confluency (<60%) and replenished with growth media containing freshly added serum daily. Cells were authenticated by measuring the myogenic potential (%MyoG+ cells and myofiber fusion index after 3 and 7 days of differentiation). New vials of cells were started every 4 weeks and tested for myogenic potential, as described above. Cells were cultured in DMEM (Gibco) with 10% fetal bovine serum (FBS) and 1% penicillin/streptomycin (pen-strep) at 37°C with 5% $CO_2$. Cells were passaged every 2 days. For glutamine and serum withdrawal experiments, cells were washed once with 1× phosphate buffer saline (PBS) and cultured for 12 hr in DMEM lacking glutamine or serum (Gibco), before re-stimulation with glutamine or serum for the time period indicated. For differentiation experiments, DMEM containing 2% horse serum and 1% pen-strep was used to induce differentiation of C2C12 mouse myoblasts. The differentiation medium (DMEM with 2% horse serum) was refreshed every 24 hr. For stable expression in 293T and C2C12 cells, puromycin selection

proceeded for 2 weeks at a concentration of 2 µg/ml. Cells were counted using the Countess 3 Automated Cell Counter (Thermo Fisher).

## mESC culture

mESCs were previously generated from C57BL/6x129 F1 male embryos (*Banaszynski et al., 2013*). mESCs were maintained on gelatin-coated plates in 2i/LIF medium containing a 1:1 mix of DMEM/F-12 (Gibco) and Neurobasal medium (Gibco) including N-2 supplement (Gibco), B-27 supplement (Gibco), 2-mercaptoethanol, 2 mM Glutamax (Gibco), LIF (produced in house) supplemented with 3 µM CHIR99021 (Stemgent), and 1 µM PD0325901 (Stemgent) (2i). When assessing the effect of PASKi on mESCs, the 2i supplement was replaced with 75 µM of BioE-1197+LIF (PASKi).

## EB formation assay

For EB formation, mESCs were diluted to $10^4$ cells/ml in EB differentiation media (DMEM, 15% FBS, 1× MEM-NEAA, 1× pen-strep, 50 µM β-mercaptoethanol) and 30 µl drops were placed on the lid of a 150 mm dish. The lid was inverted and placed over a dish containing 10–15 ml of PBS. The hanging drops were cultured for 3 days at 37°C and 5% $CO_2$. The hanging drops were then washed from the lids with EB differentiation media and cultured in 100 mm dishes on an orbital shaker at 50 rpm for an additional day.

## Preparation of protein lysates

293T cells or C2C12 cells were seeded in 100 mm dishes overnight prior to transfection using 1:3 ratio of DNA to JetPrime reagent (Polyplus) in JetPrime Buffer (Polyplus) at approximately 80% confluency, according to the manufacturer's protocol. Cells were scraped and lysed 24 hr following transfection in native cell lysis buffer (20 mM Tris-HCl pH 7.5, 150 mM NaCl, 1 mM EDTA, 1 mM NaF, 1 mM beta-glycerophosphate, 1% Triton X-100) containing freshly added protease and phosphatase inhibitors (1 mM PMSF, 1× protease inhibitor cocktail, 1× phosphatase inhibitor cocktail; Sigma) for 20 min on ice. For denaturing lysis to prepare samples for acetyl-lysine pulldown, homogenized tissue samples or cells were lysed in RIPA lysis buffer (25 mM Tris pH 7.6, 150 mM NaCl, 1% NP-40, 1% sodium deoxycholate, 0.1% SDS) containing freshly added protease and phosphatase inhibitors (1 mM PMSF, 1× protease inhibitor cocktail, 1× phosphatase inhibitor cocktail; Sigma). Whole-cell lysates were cleared by centrifugation at 15,000 rpm for 20 min. Protein quantification was performed using BCA kit (Pierce), following the manufacturer's instructions.

## Co-immunoprecipitation

Cell lysates were prepared as described above. 10% of each lysate was collected (input) and the remaining lysate was incubated with Anti-V5 Agarose Affinity Gel beads (Sigma) or Anti-FLAG M2 magnetic beads (Sigma) overnight at 4°C. IP samples were washed the following day with wash buffer (20 mM Tris-HCl pH 7.5, 150 mM NaCl, 1% Triton X-100) on ice for five times each. For acetyl-lysine pulldowns, IP samples were washed with RIPA lysis buffer (25 mM Tris pH 7.6, 150 mM NaCl, 1% NP-40, 1% sodium deoxycholate, 0.1% SDS). Samples were resolved via SDS-PAGE and immunoblot analysis was conducted with the antibodies listed.

## Subcellular fractionation

For subcellular fractionation, C2C12 cells were seeded in 100 mm dishes and scraped directly into PBS. Cells were subsequently centrifuged at 1000 rpm for 2 min at 4°C. The pellet was washed with additional PBS twice (1000 rpm for 2 min for each wash), before the pellet was resuspended in 200 µl of ice-cold fractionation buffer (10 mM HEPES, 10 mM KCl, 1.5 mM $MgCl_2$, 0.34 mM sucrose, 10% glycerol, 1 mM DTT, 1 mM PMSF, protease inhibitor cocktail). Triton X-100 was subsequently added to 0.1% final concentration. The lysate was incubated on ice for 8 min, before centrifugation at 1300×$g$ for 5 min at 4°C. The supernatant fraction was collected, while the pellet washed twice in additional fractionation buffer containing 0.1% Triton. The collected supernatant fraction was centrifuged at 20,000×$g$ for 5 min at 4°C (collecting the supernatant as the cytosolic fraction). The pellet (nuclear fraction) was resuspended in lysis buffer containing 20 mM HEPES pH 7.9, 1.5 mM $MgCl_2$, 0.5 M NaCl,

0.2 mM EDTA, 20% glycerol, 1% Triton X-100, 1 mM PMSF, protease inhibitor cocktail for 10 min at 4°C. The nuclear fraction was clarified via centrifugation at 20,000×$g$ for 5 min at 4°C.

## Isolation of primary myoblasts

Primary myoblasts were derived using FACS- or MACS-based sorting using published protocol or pronase-based myoblast isolation protocol. All cells were tested for contaminating non-MuSCs cells (e.g. fibroblasts) to ensure the purity of cell preparation. Furthermore, we compared the emergence of progressive heterogeneity in Pax7+ MuSCs cultures using three different methods of isolation from $Pask^{WT}$ and $Pask^{KO}$ animals. These ensured genotype-specific effects on MuSCs heterogeneity in culture, and not due to the method used to isolate MuSCs.

The following method was used for FACS- and MACS-based MuSCs isolation to obtain cell suspension from hindlimb muscle fragments. TA, gastrocnemius, soleus, and extensor digitorum longus were harvested from the hindlimbs of two mice and were pooled prior to mincing and digestion in 10 ml of digestion enzyme cocktail containing 800 U Collagenase B (Roche 11088831001) and 1.1 U Dispase II (Roche 04942078001). The tissue and enzyme mixture was incubated in a 37°C for 1 hr water bath with gentle agitation. Following digestion, tissue slurry was triturated and filtered using a 100 µm filter. Dispersed cells were collected by centrifugation at 500×$g$ for 5 min at 4°C. Cell pellets obtained by this method were further subjected to purification either using FACS- or MACS-based methods as follows.

### Magnetic-activated cell sorting

For MACS, the Satellite Cell Isolation Kit (Miltenyi Biotec, Cat# 130-104-268) was used to enrich CD31$^-$, Sca-1$^-$, and CD45$^-$ cells using LS column attached to QuadroMACS separators according to the manufacturer's protocol. The resulting flow-through was captured and incubated with anti-α7-Integrin microbeads (Miltenyi Biotec, Cat# 130-104-261) for 20 min on ice to obtain a relatively pure population of quiescent satellite cells. After incubation, the cell suspension was fed to a new LS column attached to the QuadroMACS separator, and unbound cells were allowed to flow through the column. Next, the column was washed two times with a wash buffer (1× PBS without Ca$^{2+}$ and Mg$^{2+}$, 5% BSA, 2 mM EDTA). Finally, cells were released from the column and plated on ECM hydrogel or Matrigel-coated glass coverslips in 24-well plates in myoblast proliferation media (DMEM high glucose with sodium pyruvate and glutamine, 20% FBS (vol/vol), 1% pen-strep (vol/vol) and 2.5 ng/ml bFGF).

### Fluorescence-activated cell sorting

FACS-based isolation of MuSCs was performed as described previously with minor modifications (*Liu et al., 2015*). The cell pellet obtained after enzymatic digestion described above was resuspended with 5 ml of FACS buffer (1× PBS without Ca$^{2+}$ and Mg$^{2+}$, 5% BSA, 2 mM EDTA) and spun down at 500×$g$ × 5 min at 4°C. The resulting pellet was resuspended in 1 ml FACS buffer containing FITC-conjugated anti-CD31, anti-CD45, anti-Sca-1, and biotin-conjugated anti-VCAM1, 647-conjugated anti-α7-Integrin antibodies and incubated at 4°C for 60 min. After incubation, cells were spun down at 500×$g$×5 min at 4°C. The cell pellet was washed 2× with FACS buffer, resuspended in 1 ml of FACS buffer containing PE-Cy7-streptavidin, and incubated at 4°C for 60 min. Following incubation, cells were washed with FACS buffer and filtered through a 35 µM strainer cap into a 5 ml polypropylene round-bottom FACS tube. Samples were sorted at the University of Kentucky Flow Cytometry and Immune Monitoring Core Facility.

### Pronase-based primary myoblasts isolation

Myoblasts were isolated using the pronase-based method as described previously (*Danoviz and Yablonka-Reuveni, 2012*). Briefly, TA muscles from the hindlimbs of mice were isolated, minced in DMEM, and enzymatically digested with 0.1% pronase for 1 hr. After repeated trituration, the cell suspension was filtered through a 100 µM filter. Cells were plated on Matrigel-precoated plates in myoblast growth media as above.

## Treatments

For PASKi treatment, 50 µM PASKi or vol/vol DMSO was added to myoblast proliferation media while plating freshly isolated MuSCs, which were replaced every 24 hr until the duration of the experiment.

For glutamine withdrawal experiments, DMEM without glutamine (Thermo Fisher, 10313021) containing 10% FBS, 2.5 ng/µl bFGF, 1% pen-strep was used 24 hr after MuSCs activation and proliferation in standard myoblast proliferation media. Control cells (+Gln) were kept in the same media but containing exogenously added 2 mM glutamine.

## Cell synchronization

Cell synchronization was performed as described previously with minor modifications (*Kikani et al., 2012*). For cell synchronization, cells were plated at $5 \times 10^5$ cells per 100 mm dish containing DMEM supplemented with serum and antibiotics. After 24 hr, growth media was replaced with media containing 5% serum with aphidicolin (2 µg/ml) for 12 hr, and then with media containing 1% serum with aphidicolin (2 µg/ml) for 8 hr to synchronize the cells in the G1 phase. To arrest cells in the G2/M phase, cells were released in media containing DMEM+10% FBS+1 % PS+2 µM nocodazole for 24 hr. M-phase arrested cells were collected by trituration followed by centrifugation at 1500 rpm × 5 min at 4°C. Cell pellets were prepared as described above to perform IP experiments. For testing the effect of glutamine in mitotic cells, aphidicolin arrested cells were washed twice with media lacking glutamine prior to releasing cells in DMEM without glutamine+10% FBS+1% PS+2 µM nocodazole.

## SDS-PAGE and immunoblotting

Whole-cell lysates (15 µg) or IP samples were denatured at 95°C for 5 min before being loaded into 10% SDS-PAGE gels submerged in running buffer (25 mM Tris, 192 mM glycine, 0.1% SDS). A constant voltage of 130 V was applied for 1.5 hr. Protein samples were transferred to nitrocellulose membranes (0.45 µm) for 1.5 hr at 100 V (constant voltage) in 1× Towbin Buffer (25 mM Tris, 192 mM glycine, 20% [vol/vol] methanol, 0.05% sodium dodecyl sulfate pH 8.3). Membranes were blocked in 3% fish gelatin (Sigma) dissolved in TBS-T (20 mM Tris pH 7.5, 150 mM NaCl, 0.1% Tween-20) for 1 hr prior to incubation with the primary antibodies specified overnight at 4°C. Following incubation, membranes were washed three times with TBS-T prior to immunoblotting with the HRP-linked or fluorescent secondary antibodies for 1 hr at room temperature. Blots were then washed three times with TBS-T and incubated with enhanced chemiluminescent (ECL) substrate (Clarity, Bio-Rad) for 1 min to detect HRP-linked antibodies or washed in PBS for 5 min (for fluorescent antibody detection). ChemiDoc MP (Bio-Rad) was used for ECL detection and Odyssey CLX (LI-COR) was used for fluorescent detection at near-IR wavelengths of 680 nm or 800 nm.

## Immunofluorescence and immunohistochemistry

For all immunofluorescence microscopy, cells were seeded in 24-well plates on glass coverslips precoated with 0.1% gelatin (HEK293T or C2C12 cells) or 1 mg/ml Matrigel for primary cells. At the end of all experimental time-points, cells were fixed with 4% paraformaldehyde (PFA, EM grade, Electron Microscopy Services) in 1× PBS for 15 min before wells were washed three times with 1× PBS. Samples were subsequently permeabilized with 0.2% Triton X-100 in 1× PBS for 10 min at room temperature before blocking in 10% normal goat serum in 1× PBS for 1 hr. Primary antibodies indicated were diluted in 1× PBS and subsequently added to all wells. Plates were incubated overnight in a humidified chamber at 4°C, before wells were washed with 1× PBS three times. Secondary antibodies (Alexa Fluor, Thermo Fisher) were then added to the wells, and incubation proceeded for 1 hr at room temperature. The wells were washed three times with 1× PBS, before coverslips were mounted with ProLong Diamond Antifade mounting media with DAPI (Thermo Fisher) onto glass slides. After coverslips were allowed to cure overnight at room temperature, the slides were imaged with confocal microscopy (Nikon A1R). Images were analyzed and quantified using calibrated Fiji software.

## Immunohistochemistry for Pax7 staining

For detection of Pax7 from the tissue sections, freshly isolated muscles at indicated time-points after muscle injury were fresh-frozen in OCT in 2-methylbutane. Muscles were cross-sectioned at 10 µm, air-dried for 20 min, and fixed for 20 min in 4% PFA in PBS. Sections were subjected to antigen retrieval using a 2100 PickCell Retriever followed by quenching for 5 min in 3% $H_2O_2$. Tissue sections 60 min in 5% normal goat seum, followed by mouse-on-mouse block for 30 min. Sections were incubated overnight at 4°C in appropriate primary antibody, washed three times in PBS and incubated for 2 hr at room temperature in secondary antibody. Sections were washed in PBS and Tyramide signal

amplification (Biotium) was performed according to the manufacturer's instructions. A glass coverslip was mounted using Prolong Anti-Fade reagent with DAPI.

## RNAseq methods and analysis

Total RNA was isolated from cultured C2C12 myoblasts using a RNeasy Mini Kit (QIAGEN). Agilent 2100 TapeStation was used to determine RNA quality and samples with RNA Integrity Numbers (RIN) 8 were chosen for RNAseq. Libraries were prepared using TruSeq Stranded Total RNA Library Prep Kit with Ribo-Zero Gold (Illumina). Sequencing was performed on the Illumina Hiseq 2500 with Paired-End reads. De-multiplexed read sequences were aligned to the *Mus musculus* mm10 reference sequence using TopHat2 splice junction mapper. Raw counts were normalized, and differentially expressed genes were called using DESeq2 analysis. Heatmaps, volcano plots, and hallmark pathway analysis were generated by R. Gene set enrichment analysis (*Mootha et al., 2003*; *Subramanian et al., 2005*) was performed using the hallmark gene sets (MSigDB hallmark gene sets) and a previously identified gene signature for stemness (*Wong et al., 2008*).

## Quantitative real-time PCR

Total RNA was extracted using the RNeasy Kit (QIAGEN) after all treatments are accomplished according to the protocol supplied by the manufacturer. RNA was quantified using a Implen MP80 and 1 µg of RNA was used for reverse transcription (QuantiTect Reverse Transcription Kit). cDNA samples were diluted 1:16 and used for real-time qPCR (Applied Biosystems) using Power Up SYBR Green PCR Master Mix (Applied Biosystems). A standard curve was prepared to obtain relative quantity for each experimental primer sets, which were normalized with 18s rRNA levels.

## Muscle injury and regeneration

Muscle injury and regeneration animal experiments were performed in accordance with protocols approved by the Institutional Animal Care and Use Committee at the University of Kentucky to CK (2019-3317). For muscle injury, 1.2% $BaCl_2$ was freshly prepared in sterile distilled water, and 25 µl was injected intramuscularly into TA muscles of 4- to 8-month-old anesthetized mice. For BPTES or DMSO injection, 10 µM working stock of BPTES was prepared in sterile 0.1× PBS. 10 µl of 10 µM BPTES or 0.001% (vol/vol) DMSO in 0.1× PBS was intramuscularly injected into TA muscles 5 days after the $BaCl_2$-induced injury every 24 hr for 3 days. Animals were monitored to ensure full recovery from anesthesia and followed for a duration set forth by experimentation.

## Metabolomics sample preparation

$3\times10^5$ cells/well with five replicates per condition were seeded in six-well plates 1 day before metabolite extraction. An identical cell counting plate was seeded for each condition. Cell culture media was aspirated thoroughly from cell culture wells and rapidly washed twice with 10 ml and 5 ml of 0.1× PBS and subsequently placed on ice after PBS washes. Metabolite extraction buffer was added to each well (1000 µl of 50% methanol with 20 µM L-norvaline [as an internal control]) on ice and plates were then transferred to –20°C for 10 min. Cells were then scraped with a cell scraper (Sarstedt) and the entire volume of each well was transferred to 1.5 ml Eppendorf tubes on ice. Samples were then thoroughly homogenized on a disruptor genie (Scientific Industries) at 3000 rpm for 3 min at room temperature. After homogenization, samples were then centrifuged at 15,000 rpm for 10 min at 4°C. The top 90% of sample supernatant was transferred to 1.5 ml tubes and samples were stored at –80°C until GC-MS metabolite quantification. Once the supernatant was removed, the insoluble pellet was washed four times with 50% methanol before a final wash of 100% methanol. Between each wash, pellets were homogenized on a disruptor genie at 3000 rpm for 1 min and then spun down at 15,000 rpm at 4°C. Following washes, the pellet was hydrolyzed in 3 N HCl for 2 hr at 95°C on a shaking Thermomixer (Eppendorf). The reaction was quenched with 100% methanol containing 40 µM L-norvaline (as an internal control). The sample was then incubated on ice for at least 30 min. The supernatant was collected by centrifugation at 15,000 rpm at 4°C for 10 min and both polar and hydrolyzed samples subsequently dried by vacuum centrifuge at $10^{-3}$ mbar (*Andres et al., 2020*; *Sun et al., 2019*).

## Metabolomics analysis

Dried polar and hydrolyzed samples were derivatized by the addition of 20 mg/ml methoxyamine hydrochloride in pyridine and incubation for 1.5 hr at 30°C. Sequential addition of

*N*-methyl-trimethylsilyl-trifluoroacetamide (MSTFA) followed with an incubation time of 30 min at 37°C with thorough mixing between the addition of solvents. The mixture was then transferred to a V-shaped amber glass chromatography vial. An Agilent 7800B gas chromatography (GC) coupled to a 7010A triple quadrupole mass spectrometry (MS) detector equipped with a high-efficiency source was used for this study. GC-MS protocols were similar to those described previously (*Sun et al., 2019*; *Andres et al., 2020*), except a modified temperature gradient was used for GC: Initial temperature was 130°C, held for 4 min, rising at 6 °C/min to 243°C, rising at 60 °C/min to 280°C, held for 2 min. The electron ionization (EI) energy was set to 70 eV. Scan (*m/z*: 50–800) and full scan mode were used for target metabolite analysis. Metabolite EI fragmentation pattern and retention time were determined by ultrapure standard purchased from Sigma. Ions (*m/z*) and retention time (min) used for glycogen quantitation was glucose (160 or 319 *m/z*; 17.4 min). Relative abundance was corrected for recovery using the L-norvaline standard and adjusted to protein input represented by pooling amino acids detected by GC-MS (*Andres et al., 2020*; *Sun et al., 2019*).

## Quantification and statistical analysis

A two-tailed t-test was performed in GraphPad Prism 9 to determine significant differences between the relevant sample and control groups (indicated in the Results). *, **, *** on graphs represent $p < 0.05$, $p < 0.01$, $p < 0.001$, respectively. A two-tailed t-test was used to compare differences. Error bars are represented as the standard deviation. All western blot experiments were done at least three times under identical conditions and best data are represented. For all microscopy experiments, three independent wells were analyzed and at least five fields from each well were quantified. We include images with multiple cells in the field to provide accurate comparison of analyzed data and to ensure images represent the quantification.

## Resource availability

Further information and requests for reagents and resources should be directed to the lead contact, Chintan Kikani (chintan.kikani@uky.edu).

## Materials availability

Plasmids generated in this work are available upon request (*Supplementary file 1*). Please direct inquiries to the lead contact. Other reagents used are commercially available.

## Acknowledgements

We acknowledge Jared Rutter and Wojciech Swiatek for providing a PASK inhibitor. In addition, we thank Cory Dungan (Center for Muscle Biology, University of Kentucky) and Kevin Murach (University of Arkansas Medical Center) for stimulating discussion and providing resources, reagents, and training in performing muscle histology. Funding This work was supported by funding from the National Institute of Arthritis and Musculoskeletal and Skin Diseases (NIAMS), 1R01AR073906-01A1, and the College of Arts and Science start-up support to CKK. RCS is supported through NIH R01 grant AG066653-01; St. Baldrick's career development award; Rally foundation independent investigator grant; V-scholar foundation award; and University of Kentucky College of Medicine and Markey Cancer Center start-up funds, and from the National Cancer Institute and NIH/NCI F99CA264165 (LEAY). LAB is supported by the National Institute of General Medical Sciences (NIGMS) GM124958, Welch Foundation I-2025, and American Cancer Society (134230-RSG-20-043-01-DMC) and to SM (fellowship support from UT Southwestern Medical Center Hamon Center for Regenerative Sciences and Medicine).

# Additional information

## Funding

| Funder | Grant reference number | Author |
|---|---|---|
| National Institute of Arthritis and Musculoskeletal and Skin Diseases | 1R01AR073906-01A1 | Chintan K Kikani |
| National Cancer Institute | AG066653-01 | Ramon C Sun |
| National Institute of General Medical Sciences | GM124958 | Laura A Banaszynski |
| National Cancer Institute | F99CA264165 | Lyndsay EA Young |
| American Cancer Society | 134230-RSG-20-043-01 | Laura A Banaszynski |
| Eunice Kennedy Shriver National Institute of Child Health and Human Development | R01 HD109239 | Laura A Banaszynski |

The funders had no role in study design, data collection and interpretation, or the decision to submit the work for publication.

## Author contributions

Michael Xiao, Data curation, Formal analysis, Validation, Visualization, Methodology, Writing – original draft, Writing – review and editing; Chia-Hua Wu, Data curation, Supervision, Validation, Investigation, Visualization, Writing – original draft; Graham Meek, Data curation, Formal analysis, Validation, Visualization, Methodology, Project administration; Brian Kelly, Data curation, Investigation, Methodology; Dara Buendia Castillo, Data curation, Validation, Writing – original draft; Lyndsay EA Young, Data curation, Software, Methodology; Sara Martire, Data curation, Validation, Methodology; Sajina Dhungel, Validation, Investigation; Elizabeth McCauley, Investigation; Purbita Saha, Altair L Dube, Data curation, Validation; Matthew S Gentry, Supervision, Funding acquisition, Writing – original draft; Laura A Banaszynski, Conceptualization, Funding acquisition, Validation, Investigation, Writing – original draft, Project administration, Data curation, Formal analysis, Methodology, Resources, Supervision, Writing – review and editing; Ramon C Sun, Supervision, Funding acquisition, Investigation, Writing – original draft; Chintan K Kikani, Conceptualization, Resources, Data curation, Software, Formal analysis, Supervision, Funding acquisition, Validation, Investigation, Visualization, Methodology, Writing – original draft, Project administration, Writing – review and editing

## Author ORCIDs

Chia-Hua Wu http://orcid.org/0000-0002-5361-5469
Dara Buendia Castillo http://orcid.org/0000-0001-8955-2984
Matthew S Gentry http://orcid.org/0000-0001-5253-9049
Ramon C Sun http://orcid.org/0000-0002-3009-1850
Chintan K Kikani http://orcid.org/0000-0003-1140-0192

## Ethics

Muscle Injury and Regeneration Animal experiments were performed in accordance with protocols approved by the Institutional Animal Care and Use Committee at the University of Kentucky to CK (2019-3317).

## Decision letter and Author response

Decision letter https://doi.org/10.7554/eLife.81717.sa1
Author response https://doi.org/10.7554/eLife.81717.sa2

# Additional files

## Supplementary files
• MDAR checklist

• Supplementary file 1. Plasmids available upon request.

## Data availability
RNA sequencing gene expression and other source data are provided along with this manuscript.

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
