## [Editor Report]

The study by Xiao et al. presents an important finding in the area of metabolic regulation underpinning cell fate decisions in murine muscle stem cells. Combining multiple approaches, the study provides convincing evidence of glutamine-dependent control of the sub-cellular localisation of the kinase PASK and the consequent activation of myogenic programs. The study will be of interest to researchers in the areas of stem cells, regeneration, and metabolic signalling.

---

## [Decision Letter]

**Decision letter after peer review:**

Thank you for submitting your article "PASK relays metabolic signals to mitotic Wdr5-APC/C complex to drive exit from self-renewal" for consideration by *eLife*. Your article has been reviewed by 3 peer reviewers, and the evaluation has been overseen by a Reviewing Editor and Marianne Bronner as the Senior Editor. The following individual involved in the review of your submission has agreed to reveal their identity: Apurva Sarin (Reviewer #1).

Essential revisions:

The core ideas of the manuscript by Xiao et al., are of interest. However, substantial issues – outlined below – remain to be addressed. Suggestions for experiments and analysis are provided. Key concerns relate to the purity of primary cells isolated from mice, arising from the method adopted for isolation and subsequent characterisation. This section of the work requires fresh experiments (summarised in #1& 2) to be performed, with careful attention to methodology. Characterization of the mouse model with the genetic ablation of PASK is also required.

1. The method by which cells are isolated is yielding a heterogenous cell population, which is likely contaminated by fibroblasts. While pre-plating can be used to isolate muscle stem cells and culture them as myoblasts, it takes days of growth and multiple rounds of passaging in order to get a more pure population of myogenic cells. This would account for the high number of Pax7 negative cells in the primary myoblast experiments (~50% in some conditions) as they are most likely fibroblasts, which can be confirmed by staining for fibroblast markers. The increase in Pax7 cells in certain conditions could also simply be due to the loss of contaminating cell types due to the treatment. The experiments using myoblasts must be redone using a more appropriate cell isolation method (i.e. FACS-based analysis) or by culturing these isolated cells for a much longer period of time to eventually get a more pure cell population as confirmed by analysis of specific markers.

2. The characterization of the PASK knockout model is incomplete and a characterization is required. Basic muscle functional experiments such as EDU incorporation for proliferation and fusion index for differentiation should be provided in WT and KO cells to assess the effects of the loss of PASK. Further, since the PASK KO can regenerate muscle post-injury, it suggests that PASK is not necessary for the establishment of the progenitor population. The text may be modified to indicate this. Isolating EDL myofibers and stain for PASK and PAX7 at 0, 24, 48, and 72-hour post isolation will allow tracking changes in PASK expression and cell localization, as well as confirm the number of muscle stem cells in WT and KO mice, during quiescence and during the process of muscle stem cell activation, proliferation and differentiation in a near in vivo context. In general, performing repeated intramuscular injections into regenerating muscle is not recommended as the injections will re-damage the muscle. If this could be avoided and injected elsewhere that would be preferable.

3. In the data shown in Supp 2A western blot of PASK a significant fraction of PASK is localized in the nucleus. Can the author explain the discrepancies?

4. In figure 4, the author showed that upon glutamine addition, it was sufficient for PASK to be acetylated. However, would the author clarify if experiments have been performed in glutamine deficient medium? It would be helpful to provide data for serum depletion conditions as well. In Supp 4D, authors claim that a significant fraction of mouse PASK is acetylated during regeneration on days 3 and 5 post-injury. However, the bands are weak with a lot of noise and don't look convincing. Can they explain the huge noise in such blots?

5. In Figure 1A, Rex1 is described as being significantly higher in the PASKi condition, yet they look very similar. Figure 1C legend says that the cells were treated for 4 days but the images are only up to 48 hours. An image of the later time point may be included with staining for MyHC or myogenin.

6. In Figure 2A individual channels may be shown. In Figure 2B, the data are not internally consistent. The DAPI intensity, which should be a control is highly variable. This whole panel could be removed and better visualized using Panel A with individual channels. In Figure 2C, the DAPI channel would be a good addition In Figure 2D, the Y-axis is inappropriate. The data could be shown in two graphs, with % of MyoG+ cells, plotted separately from % of PASK localization / Myog+ cells. The events of exclusion of Pax7 from daughter mitotic cells with asymmetric nuclear localization of PASK in 2G should be quantified and presented as a bar graph. Further, it would be important to stain for a cell membrane protein to definitively show that this is a dividing cell instead of just 2 cells that are very close to one another.

7. Can the authors elaborate on the data in 4B? It is interesting that the acetylation of PASK is transient. Does this mean that the acetylation is only necessary for the PASK to enter the nucleus but afterwards is deacetylated? Or are these cells only getting nuclear PASK very transiently and then the PASK returns to the cytoplasm? Fractionation of these cells to see if the nuclear PASK is still acetylated would be interesting. Figure 4 D+I, the control does not have an error like the other conditions do.

8. In Figure 5F does the Y-Axis mean the number of double positive Pax7 and Brdu cells, or does it mean % of Pax7+ cells compared to BrdU^+^ cells? The data may be shown as % of double positive Pax7/BrdU cells over all cells for greater clarity. Similarly, panel I in the same figure may be replotted such that the y-axis is # Pax7+ cells/mm2, as the observer's field of view is not known. In panels H^+^I, the number of satellite cells in the DMSO condition appears to be extremely low, especially for an injured muscle.

9. The legend to Figure 6D states that the scale bar is 20 um, but it's a bar graph and has no scale bar. The y-axis should be Pax7 cells/mm2. Data on the control siRNA and percent knock-down is missing in the experiment shown in 6G-H. A qPCR needs to be done to establish the efficacy of siRNA treatments, or alternatively, a 3 colour stain of GFP, PASK, and PAX7 should be performed. Figure 6H^+^J the y-axis should be % of Pax7+ cells.

10. In Figure 7, mTOR phosphorylation is presented as a requirement for PASK association with Wdr5. While experiments show that glutamine tunes this association, it would be important to understand if PASK acetylation directly impacts PASK- Wdr5 interactions. Further, The legend in this figure skips J and goes directly to K.

General comments

11. In numerous figure panels, the y-axis represents the # of cells, rather than a percentage or ratio. This is uninformative as the number of cells will never be the same between conditions and experiments. Please replace these panels with a more appropriate y-axis (each will be identified individually in the minor concern section).

12. Indicating molecular weight markers and showing complete immunoblots (particularly in the case of immunoprecipitation) would be helpful. Please state the number of experiments (n) performed for all IP blot and Western blot experiments in legends.

13. Listing the number of replicates and types of replicates(biological/technical) performed for each experiment (for both blot data and imaging experiments) would be helpful.

14. In multiple legends it says that the error bar represents SD, but in the statistical methods, it says that error bars are SEM. SD is preferable.

15. Line 344-348 refers to C2C12 myoblasts as Pax7+ stem cells. C2C12s are not stem cells and should never be labelled as such.

*Reviewer #1 (Recommendations for the authors):*

Listed below are some points of concern for the authors' consideration

1. Indicating molecular weight markers and showing complete immunoblots (particularly in the case of immunoprecipitation) is required for better visualization of the data.

2. There is a mismatch in the description of the text (lines 97-101) and the figure/ extended legends.

3. Which construct does WT NLS-hPASK recombinant (line 196) refer to? A table listing the mutants generated in the study would be a useful reference for a reader.

4. Data on the control siRNA is missing in the experiment shown in Figure 6G-H.

5. The analysis of the sequences regulating PASK functions is quite comprehensive. However, the amino-acid residue(s) that are targets of acetylation are not identified. Are these known?

6. In Figure 7, mTOR phosphorylation is presented as a requirement for PASK association with Wdr5. While experiments show that glutamine tunes this association, it would be important to understand if PASK acetylation directly impacts PASK- Wdr5 interactions.

7. Confirming glutamine regulation of PASK-Wdr5 interactions in primary myoblasts (endogenous proteins) will strengthen the conclusions of this otherwise detailed and comprehensive study.

8. Pharmacological inhibitors have been relied on for most experiments. In this context, verifying key observations using additional specific inhibitors (with specificity controls) and/or gene silencing should be considered. GLS1 and CBP/p300 can be readily tested using siRNA in the experimental systems in the study.

*Reviewer #2 (Recommendations for the authors):*

1) Supp 2A western blot of PASK doesn't show convincing data that a significant fraction of PASK is localized in the nucleus. Can the author explain the discrepancies?

2) Can authors also quantify the events of exclusion of Pax7 from daughter mitotic cells with asymmetric nuclear localization of PASK in 2G and present it in a bar graph?

3) In Supp 4D, authors claim that a significant fraction of mouse PASK is acetylated during regeneration on days 3 and 5 post-injury. However, the bands are weak with a lot of noise and don't look convincing. Can they explain the huge noise in such blots?

4) Can the authors also add the number of experiments (n=?) they have performed for all IP blot and Western blot experiments?

5) The authors can improve on the current paper by listing the number of replicates and types of replicates(biological/technical) performed for each experiment (for both blot data and imaging experiments).

6) In figure 4, the author showed that upon glutamine addition, it was sufficient for PASK to be acetylated. However, would the author clarify if experiments have been performed in glutamine deficient medium?

7) When they did the experiment where they looked at the acetylation level by western blot (figure 4). It would be really helpful to have that for serum depletion conditions as well.

*Reviewer #3 (Recommendations for the authors):*

The authors have underutilized the mouse model of Pask.

The authors should follow standard FACS protocol for high-purity isolation of muscle stem cells. their current method will likely introduce a vast amount of contamination of fibroblasts which can make the conclusion drawn from the study questionable.

[Editors' note: further revisions were suggested prior to acceptance, as described below.]

Thank you for resubmitting your work entitled "PASK links cellular energy metabolism with a mitotic self-renewal network to establish differentiation competence" for further consideration by *eLife*. Your revised article has been evaluated by Marianne Bronner (Senior Editor) and a Reviewing Editor.

The manuscript has been improved but there are some remaining issues that need to be addressed, as outlined below:

The reviewers agree that the authors have addressed many of the issues raised by the reviewers. Notably, fibroblast contamination has been addressed in the resubmission using different protocols for the purification of myoblasts.

While the work remains of interest, the use of myoblasts with low levels (30% or less) Pax7+ in the analysis (to verify and extend observations made in C2C12 cells), remains a significant concern. This has remained unaddressed in the revised manuscript. The loss of Pax7 expression in myoblasts on culture is unexpected and discordant with current observations in the field and raises concerns about the myogenic status of the cells employed. Confirmation of key observations in demonstrably Pax7-positive myoblasts is essential for concerns to be comprehensively addressed.

Additional comments.

– WT Pax7 levels do not match in the image and plot shown in Figure 1 panels K-L.

– Lines 205-208 in the results probably refer to data shown in Figure 2 and not Figure 1.

– In Figure 5A, the legend to the colour-coded trace is missing.

– The age of mice used in the study is to be indicated.

---

## [Author Response]

Essential revisions:The core ideas of the manuscript by Xiao et al., are of interest. However, substantial issues – outlined below – remain to be addressed. Suggestions for experiments and analysis are provided. Key concerns relate to the purity of primary cells isolated from mice, arising from the method adopted for isolation and subsequent characterisation. This section of the work requires fresh experiments (summarised in #1& 2) to be performed, with careful attention to methodology. Characterization of the mouse model with the genetic ablation of PASK is also required.1. The method by which cells are isolated is yielding a heterogenous cell population, which is likely contaminated by fibroblasts. While pre-plating can be used to isolate muscle stem cells and culture them as myoblasts, it takes days of growth and multiple rounds of passaging in order to get a more pure population of myogenic cells. This would account for the high number of Pax7 negative cells in the primary myoblast experiments (~50% in some conditions) as they are most likely fibroblasts, which can be confirmed by staining for fibroblast markers. The increase in Pax7 cells in certain conditions could also simply be due to the loss of contaminating cell types due to the treatment. The experiments using myoblasts must be redone using a more appropriate cell isolation method (i.e. FACS-based analysis) or by culturing these isolated cells for a much longer period of time to eventually get a more pure cell population as confirmed by analysis of specific markers.

We provide supporting evidence that our cell preparation using the pronase-based method (Danoviz et al., *Methods Mol Biol. 2012; 798: 21–52*) is devoid of fibroblast contamination (Figure 1 - figure supplement 3C). Moreover, our experimental approach of control treatments, a test of cell viability, proliferation measurement, and transcriptional analysis ensured accurate measurement of cellular heterogeneity in vitro in response to biochemical manipulations of pronase-isolated cells. However, the reviewers' concerns are valid, and to improve the scientific rigor and confidence in our data; we performed several key experiments to compare and reproduce results from two muscle stem cell (MuSC) isolation methods to achieve highly pure Pax7+ MuSCs:

FACS: First, we used FACS-based sorting of CD45-, Sca1-, CD31-, VCAM+ MuSCs (Based on the protocol from Dr. Thomas Rando's lab (Liu L et al., 2014)) to isolate MuSCs from *Pask^WT^* and *Pask^KO^* animals. We subjected these cells to various treatments and growth conditions as described below and quantified cellular heterogeneity in terms of Pax7+ MuSCs numbers. See Figure 1 and Figure supplement 3 for the new data generated using FACS-based isolation and the gating strategy used to purify MuSCs.

MACS: We also used a magnetic cell sorting strategy to sort CD45^-^, Sca-1^-^, CD31^-^, and Intregin α7^+^ MuSCs. We subjected MuSCs isolated using the MACS method to various culture conditions of PASK and glutamine signaling manipulations and analyzed the percent of Pax7+ SCs number, myogenic commitment, and fusion index.

Fibroblast contamination in our isolation method: We stained mononuclear cells isolated using the pronase, FACS, and MACS methods with Pax7 and vimentin antibodies. Vimentin is a marker for fibroblasts. In addition, we used mouse embryonic fibroblasts (MEFs) derived from *Pask^WT^* and *Pask^KO^* animals as a positive control for fibroblasts. Our data clearly shows no fibroblast contamination in any method used to isolate MuSCs. Thanks to the reviewers' suggestions, our data reinforce the role of glutamine/PASK in establishing the committed progenitor population by promoting heterogeneity in Pax7 levels in myoblasts. These results are replicated regardless of the method of MuSCs isolation.Stem cell heterogeneity and Pax7+ MuSCs numbers: Our data from either of the two antibody based methods for isolating MuSCs are consistent with the original data presented in the manuscript in terms of stem cell heterogeneity upon loss of PASK or glutamine signaling. Thus, genetic loss or pharmacological inhibition of PASK or glutamine signaling reduced heterogeneity in primary myoblasts by increasing the percentage of Pax7+ stem cell numbers in FACS-sorted MuSCs.

2. The characterization of the PASK knockout model is incomplete and a characterization is required. Basic muscle functional experiments such as EDU incorporation for proliferation and fusion index for differentiation should be provided in WT and KO cells to assess the effects of the loss of PASK. Further, since the PASK KO can regenerate muscle post-injury, it suggests that PASK is not necessary for the establishment of the progenitor population. The text may be modified to indicate this. Isolating EDL myofibers and stain for PASK and PAX7 at 0, 24, 48, and 72-hour post isolation will allow tracking changes in PASK expression and cell localization, as well as confirm the number of muscle stem cells in WT and KO mice, during quiescence and during the process of muscle stem cell activation, proliferation and differentiation in a near in vivo context. In general, performing repeated intramuscular injections into regenerating muscle is not recommended as the injections will re-damage the muscle. If this could be avoided and injected elsewhere that would be preferable.

Here, we address several points made in this comment:

Regeneration defect in PASK knockout animals: We have previously shown that the whole-body knockout of PASK shows a severe regeneration defect during early myogenesis (Kikani, CK, *eLife*, 2016, reproduced here in Figure 1I in the main manuscript). During an extended regeneration time course (past day 21), we noticed two phenotypes. First, we noticed smaller regenerative fiber CSA in *Pask^KO^* mice compared with *Pask^WT^* animals (Calculated in Figure 6E). Secondly, we observed elevated percentages of Pax7+ stem cells in *Pask^KO^* animals at mid-point of regeneration, Day 14 post-regeneration, compared with WT animals (Recalculated in Figure 6D). These two phenotypes confirm abnormal progression through the muscle regeneration program when PASK is deleted. While *Pask^KO^* mice appear to have developmentally normal myogenesis, we have previously reported evidence of developmental compensation in whole-body *Pask^KO^* animals in vitro (Kikani et al., 2016). This is likely due to partial compensation of catalytic functions of PASK function by other kinases that share substrate selection preference with PASK (Kikani et al., JBC 2010). We are in the process of generating tamoxifen-inducible Pax7-specific loss or gain of PASK mouse models to generate a MuSC-specific model of acute loss of PASK; however, that is not the focus of the current paper which is investigating biochemical functions of PASK.Characterization of WT and KO stem cells. We extensively characterize the entire myogenesis program in FACS-sorted MuSCs from *Pask^WT^* or *Pask^KO^* animals in Figure 1, as suggested by the reviewer. The manuscript on hand is specifically probing the biochemical functions of PASK in driving the exit from self-renewal in response to metabolic signaling. Because PASK is not expressed in quiescent MuSCs (Figure 1 —figure supplement 2A), we do not anticipate a functional role of PASK in the initial activation of QSC. Therefore, we do not propose that PASK plays a role in maintaining the QSC state or the exit and initial activation of MuSCs following muscle injury. On the other hand, PASK is transcriptionally activated in proliferating myoblasts during regeneration (Kikani et al., *eLife* 2016) and upon isolation of MuSCs (Figure 1 - figure supplement 2A). Therefore, we specifically focus on studying the functional role of PASK signaling in *activated (proliferating)* myoblasts isolated from mice or during early regeneration. We have ongoing studies examining the precise temporal kinetics of PASK transcription regulation in Pax7+ MuSCs, and identifying its upstream transcriptional regulators*.* However, we respectfully suggest that these avenues are outside the purview of this current manuscript that explicitly explores the metabolic-signaling interplay that drives the exit from self-renewal in the activated myoblasts.

3. In the data shown in Supp 2A western blot of PASK a significant fraction of PASK is localized in the nucleus. Can the author explain the discrepancies?

The western blot in Supp 2A shows nuclear translocation of PASK during early myogenesis in comparison with proliferating myoblasts. When comparing the levels of Parp between the nuclear fractions from proliferating and differentiating cells, it is evident that despite the reduction in the loading levels of Parp, noticeably more PASK is present in the nucleus of differentiating myoblasts compared with proliferating myoblasts. Thus, our biochemical fractionation confirms the microscopic analysis of the nuclear translocation of PASK during myogenesis. In addition, we have previously reported that PASK levels decline once the myogenesis program is underway. Thus, lower levels of PASK in the nuclear fraction in differentiating cells is reflective of declining levels of PASK as myogenesis is progressing.

4. In figure 4, the author showed that upon glutamine addition, it was sufficient for PASK to be acetylated. However, would the author clarify if experiments have been performed in glutamine deficient medium? It would be helpful to provide data for serum depletion conditions as well. In Supp 4D, authors claim that a significant fraction of mouse PASK is acetylated during regeneration on days 3 and 5 post-injury. However, the bands are weak with a lot of noise and don't look convincing. Can they explain the huge noise in such blots?

For experiments where we specifically tested glutamine-dependent stimulation, cells were starved overnight in media without serum and glutamine (as indicated in Figures 4G and H). The next day, we only added glutamine (no serum was added). Separately, experiments in Figures 4B and 4C show the effect of serum addition on PASK acetylation. In those conditions, we had serum containing glutamine. Finally, data in Figure 4F shows the effect of serum stimulation when glutamine metabolism is specifically inhibited, thus confirming that glutamine in serum stimulates PASK acetylation. Thus, we provide a comprehensive analysis of the effect of serum and glutamine on PASK acetylation.

PASK acetylation during regeneration: The antibody used to immunoprecipitate the total acetylome is not of high quality for muscle tissue homogenates. Despite this, we can see acetylated PASK being purified during regeneration.

5. In Figure 1A, Rex1 is described as being significantly higher in the PASKi condition, yet they look very similar. Figure 1C legend says that the cells were treated for 4 days but the images are only up to 48 hours. An image of the later time point may be included with staining for MyHC or myogenin.

Our statistical analysis of *Rex1* mRNA levels between 2i vs. PASKi conditions shows the *p* value as less than 0.05. The data in Figure 1A is plotted as ±S.D. Thus, we concluded that *Rex1* levels are significantly elevated compared in mESCs in PASKi-treated conditions compared with the 2i condition.

6. In Figure 2A individual channels may be shown. In Figure 2B, the data are not internally consistent. The DAPI intensity, which should be a control is highly variable. This whole panel could be removed and better visualized using Panel A with individual channels. In Figure 2C, the DAPI channel would be a good addition In Figure 2D, the Y-axis is inappropriate. The data could be shown in two graphs, with % of MyoG+ cells, plotted separately from % of PASK localization / Myog+ cells. The events of exclusion of Pax7 from daughter mitotic cells with asymmetric nuclear localization of PASK in 2G should be quantified and presented as a bar graph. Further, it would be important to stain for a cell membrane protein to definitively show that this is a dividing cell instead of just 2 cells that are very close to one another.

We have significantly developed this part of the manuscript in the revised version. We have shown individual channels for Figure 2A and quantified the extent of PASK cytoplasmic vs. nuclear distribution during self-renewal and differentiation. Thanks to isolation of FACS sorted MuSCs which allowes us to begin examining PASK subcellular localization earlier than the Pronase-based method, we discovered that PASK is localized into cytoplasmic granules in proliferating, Pax7+ MuSCs. We further show that PASK localized in the cytoplasmic granules disperses during differentiation in glutamine-dependent manner (Figure 6H) when PASK is localized into the nucleus. In addition, we provide an extensive analysis showing a strong connection between the extent of PASK granular localization with Pax7 expression and a negative correlation between PASK nuclear localization and Pax7 expression. Finally, we could not add cell membrane staining for the post-mitotic cells due to technical difficulties, but we have expanded our analysis by adding more examples and quantifying the data.

7. Can the authors elaborate on the data in 4B? It is interesting that the acetylation of PASK is transient. Does this mean that the acetylation is only necessary for the PASK to enter the nucleus but afterwards is deacetylated? Or are these cells only getting nuclear PASK very transiently and then the PASK returns to the cytoplasm? Fractionation of these cells to see if the nuclear PASK is still acetylated would be interesting. Figure 4 D+I, the control does not have an error like the other conditions do.

This is an excellent forward-looking question, and we thank the reviewers for pointing this out – the kinetics of PASK acetylation and nuclear translocation are intriguing and are part of our follow-up manuscript. It is noteworthy that many protein kinases, including Akt, PDK1, Erk1/2, *JNK*, and p38MAPK, exhibit transient nucleo-cytoplasmic shuttling. We have corrected the data in Figure 4 D+I.

8. In Figure 5F does the Y-Axis mean the number of double positive Pax7 and Brdu cells, or does it mean % of Pax7+ cells compared to BrdU^+^ cells? The data may be shown as % of double positive Pax7/BrdU cells over all cells for greater clarity. Similarly, panel I in the same figure may be replotted such that the y-axis is # Pax7+ cells/mm2, as the observer's field of view is not known. In panels H^+^I, the number of satellite cells in the DMSO condition appears to be extremely low, especially for an injured muscle.

The data in Figure 5F shows the ratio of Pax7+ SCs over BrdU^+^ SCs. That analysis best reflects the notion that in glutamine-depleted conditions, non-proliferating cells exhibit increased Pax7 expression, resulting in Pax7+ SC numbers. To clarify the data, we have converted our data into % of cells, showing ratio of %Pax7+ SCs/%BrdU^+^ SCs. This analysis method has been used in other publications (Liu et al., *ELife*. 2015 Aug 27;4:e09221).

We have recalculated the data from Pax7+ SC# per field to mm2 in Figure H + I.

9. The legend to Figure 6D states that the scale bar is 20 um, but it's a bar graph and has no scale bar. The y-axis should be Pax7 cells/mm2. Data on the control siRNA and percent knock-down is missing in the experiment shown in 6G-H. A qPCR needs to be done to establish the efficacy of siRNA treatments, or alternatively, a 3 colour stain of GFP, PASK, and PAX7 should be performed. Figure 6H^+^J the y-axis should be % of Pax7+ cells.

Thank you to an eagle eye reviewer. We made copy and paste error in Figure 6D, which is now corrected. Furthermore, we have provided RT-qPCR data showing siRNA knock-down of PASK in myoblasts in Figure supplement S6E and its effect on Pax7 mRNA levels. We regret these omissions and errors, and we appreciate the peer review diligence.

10. In Figure 7, mTOR phosphorylation is presented as a requirement for PASK association with Wdr5. While experiments show that glutamine tunes this association, it would be important to understand if PASK acetylation directly impacts PASK- Wdr5 interactions. Further, The legend in this figure skips J and goes directly to K.

This is a great question. The mechanistic convergence of nutrient signaling (via mTOR) and mitochondrial signaling (via glutamine) on PASK-Wdr5 interaction is of great interest to us. To specifically test this, we have ongoing work to identify acetylation sites and how these sites affect the kinetics of nuclear translocation, PASK activation by mTOR, and its association with Wdr5 (also see the response to comment # 7). To answer this question as much as possible, we tested if endogenous PASK-Wdr5 interaction is regulated in response to glutamine stimulation under the same time course where we showed PASK acetylation and nuclear translocation. Excitingly, glutamine signaling, but not serum alone (without glutamine), strengthened the endogenous PASK-Wdr5 interaction in myoblasts, thus, biochemically connecting mitochondrial glutamine signaling to epigenetic complexes via PASK acetylation (Figure 7 - figure supplement 1B).

General comments11. In numerous figure panels, the y-axis represents the # of cells, rather than a percentage or ratio. This is uninformative as the number of cells will never be the same between conditions and experiments. Please replace these panels with a more appropriate y-axis.

We have now recalculated and converted # of cells into percentage where needed.

12. Indicating molecular weight markers and showing complete immunoblots (particularly in the case of immunoprecipitation) would be helpful. Please state the number of experiments (n) performed for all IP blot and Western blot experiments in legends.

We have now provided this information.

13. Listing the number of replicates and types of replicates(biological/technical) performed for each experiment (for both blot data and imaging experiments) would be helpful.

We have now provided this information.

14. In multiple legends it says that the error bar represents SD, but in the statistical methods, it says that error bars are SEM. SD is preferable.

We only used SD in all of our analyses. We regret that we made some editing errors when we mentioned SEM in the methods. This has since been corrected.

15. Line 344-348 refers to C2C12 myoblasts as Pax7+ stem cells. C2C12s are not stem cells and should never be labelled as such.

We agree and regret this editing error. We have now corrected this mistake.

Reviewer #1 (Recommendations for the authors):Listed below are some points of concern for the authors' consideration1. Indicating molecular weight markers and showing complete immunoblots (particularly in the case of immunoprecipitation) is required for better visualization of the data.

See above – This information is provided.

2. There is a mismatch in the description of the text (lines 97-101) and the figure/ extended legends.

We thank the reviewer for pointing this out – we have corrected the error.

3. Which construct does WT NLS-hPASK recombinant (line 196) refer to? A table listing the mutants generated in the study would be a useful reference for a reader.

This is an excellent suggestion – we have now provided the table in (Figure 3 —figure supplement 1).

4. Data on the control siRNA is missing in the experiment shown in Figure 6G-H.

We have provided siRNA silencing efficiency data.

5. The analysis of the sequences regulating PASK functions is quite comprehensive. However, the amino-acid residue(s) that are targets of acetylation are not identified. Are these known?

This is an ongoing area of research in our lab. Please also see response to comment # 7 in the essential revisions.

6. In Figure 7, mTOR phosphorylation is presented as a requirement for PASK association with Wdr5. While experiments show that glutamine tunes this association, it would be important to understand if PASK acetylation directly impacts PASK- Wdr5 interactions.

This is addressed in the essential revisions.

7. Confirming glutamine regulation of PASK-Wdr5 interactions in primary myoblasts (endogenous proteins) will strengthen the conclusions of this otherwise detailed and comprehensive study.

This is addressed in the essential revisions.

8. Pharmacological inhibitors have been relied on for most experiments. In this context, verifying key observations using additional specific inhibitors (with specificity controls) and/or gene silencing should be considered. GLS1 and CBP/p300 can be readily tested using siRNA in the experimental systems in the study.

GLS1 silencing is not well-tolerated by cells, and we see greater than 50% of cell death, as opposed to acute inhibition of GLS1 by BPTES. However, by using p300 silencing by siRNA, we show that serum-induced PASK acetylation is abolished in myoblasts (Figure 4- figure supplement 1F).

[Editors' note: further revisions were suggested prior to acceptance, as described below.]

The manuscript has been improved but there are some remaining issues that need to be addressed, as outlined below:The reviewers agree that the authors have addressed many of the issues raised by the reviewers. Notably, fibroblast contamination has been addressed in the resubmission using different protocols for the purification of myoblasts.While the work remains of interest, the use of myoblasts with low levels (30% or less) Pax7+ in the analysis (to verify and extend observations made in C2C12 cells), remains a significant concern. This has remained unaddressed in the revised manuscript. The loss of Pax7 expression in myoblasts on culture is unexpected and discordant with current observations in the field and raises concerns about the myogenic status of the cells employed. Confirmation of key observations in demonstrably Pax7-positive myoblasts is essential for concerns to be comprehensively addressed.

There is substantial literature support (discussed and provided below) that isolated primary myoblasts precipitously lose Pax7 expression at population and individual cell levels as myogenic commitment is established. Pax7+ myoblast numbers in our experimental method from control conditions are consistent with considerable evidence in the literature. For example, a relatively recent study in *PNAS* (*PNAS, 2021, 118 (13) e2021093118)* showed that Pax7 protein levels were diminished within 24 hours post-isolation (Figure1A , Zhou S, et al. 2021). Furthermore, using a novel Pax7EGFP experimental model, Foteini Mourkioti’s lab very nicely showed a progressive decline in EGFP fluorescence in culture conditions within 24 hours, with nearly complete loss by 96 hours (Figure 3B, Tichy, E.D., et al. 2018).

Our manuscript explicitly addresses the mechanistic reasons driving this loss of Pax7 expression as stem cells begin to proliferate and heterogeneity in the myoblast population emerges. In our experimental method, we culture cells for up to 96 hours in the presence of media containing bFGF, with or without PASK inhibitor, or cells isolated from *Pask^KO^* animals. We track changes in the Pax7+ population in the presence or absence of PASKi, or due to genetic loss of PASK. Thus, within the experimental time frame (48 to 96 hours, depending upon the experiment), %Pax7+ myoblasts in control animals are consistent with the literature (see Figure 1A, Zhou S, et al. 2021; Figure 3B, Tichy, E.D., et al. 2018).

To further provide evidence, as requested, that PASK inhibition results in an increased proportion of Pax7+MuSCs in demonstrably Pax7+ population, we generated MuSC-specific expression of ZsGreen using the Pax7^Cre:Rosa26ZsGreen^/+ animals (Pax7^Cre^: Pax7tm1 (Cre)Mrc/J, Jackson lab, stock: 010530, B6.Cg-Gt(ROSA)26Sortm6(CAG-ZsGreen1)Hze, stock: 007906). We isolated quiescent MuSCs from the uninjured TA muscles of these animals by sorting for ZsGreen+ cells and cultured them in standard myoblast proliferation media. In these animals, ZsGreen (GFP channel) is expressed in quiescent MuSCs and will be inherited by all progeny of Pax7+ MuSCs during in vitro culture, regardless of Pax7 expression. Using this model, we show that as MuSCs begin to proliferate, Pax7+ cell numbers decline, and the GFP+ progenitor population increases in vehicle-treated cultures. In contrast, in the presence of PASKi, the increased proportion of cells were positive for both Pax7 and GFP. Thus, catalytic inhibition of PASK preserved Pax7 expression in lineage-traceable proliferating primary myoblasts.

**Author response image 1. sa2fig1:** PASK inhibition preserves Pax7 expression in proliferative myoblasts. A. MuSCs were isolated from TA muscles of 4 months old mice using the collagenase-dispase method. GFP+ cells were sorted using flow-cytometry and cultured in myoblast proliferation media in the presence of DMSO or 50 µM PASKi. Cells were fixed at different time points, as indicated in A and B. Cells were stained using Anti-Pax7. Scale bars = 20 µm (12hr and 36hr) and 40 µm (72hrs). B. Quantification of % Pax7+ cells during in vitro culture of myoblasts in control (DMSO) vs. PASKi treated conditions. C. % Total cell population of Pax7+/GFP+ and Pax7-/GFP+ populations at 12, 36, and 72 hours post isolation of primary myoblasts in control (DMSO) vs PASKi cultured cells.

Additional comments.– WT Pax7 levels do not match in the image and plot shown in Figure 1 panels K-L.

We have reconfirmed our analysis and adjusted the intensities of the microscopy panel to bring out weaker Pax7+ SCs in WT, but is not a general representation of quantified data.

– Lines 205-208 in the results probably refer to data shown in Figure 2 and not Figure 1.

Thank you for catching that. We have corrected this.

– In Figure 5A, the legend to the colour-coded trace is missing.

We have added the legend.

– The age of mice used in the study is to be indicated.

We have provided this information in methods and material.